# Linking Climate Change Information with Crop Growing Seasons in the Northwest Ethiopian Highlands

Gashaw Bimrew Tarekegn [1], Addis A. Alaminie [2] and Sisay E. Debele [3,*]

[1] Institute of Disaster Risk Management and Food security Studies, Bahir Dar University, Bahir Dar P.O. Box 5501, Ethiopia; gashbimrew@gmail.com

[2] Faculty of Civil and Water Resources Engineering, Bahir Dar University, Bahir Dar P.O. Box 400, Ethiopia; metaddi@gmail.com

[3] Global Centre for Clean Air Research (GCARE), School of Sustainability, Civil and Environmental Engineering, Faculty of Engineering and Physical Sciences, University of Surrey, Guildford GU2 7XH, UK

\* Correspondence: sd0059@surrey.ac.uk

**Abstract:** In Ethiopia, the impacts of climate change are expected to have significant consequences for agriculture and food security. This study investigates both past (1981–2010) and future (2041–2070) climate trends and their influence on the length of the growing season (LGS) in the North-Western Ethiopian highlands. Climate observations were obtained from the National Meteorological Agency of Ethiopia, while the best performing and highest resolution models from the CMIP5 experiment and RCP6 (Coupled Models Intercomparison Project and representative concentration pathway 6) were used for the analysis. Standard statistical methods were applied to compute soil water content, evaluate climate variability and trends, and assess their impact on the length of the growing season. Maximum temperature (tasmax) and minimum temperature (tasmin) inter-annual variability anomalies show that the region has experienced cooler years than hotter years in the past. However, in the future, the coolest years are expected to decrease by −1.2 °C, while the hottest years will increase by +1.3 °C. During the major rainfall season (JJAS), the area has received an adequate amount of rainfall in the past and is very likely to receive similar rainfall in the future. On the other hand, the rainfall amount in the season February to May (FMAM) is expected to assist only with early planting. For the season October to January (ONDJ), the rainfall amount may help lengthen the growing season of JJAS if properly utilized; otherwise, the season has the potential to destroy crops before and during the harvesting time. The soil water content changes in the future remain close to those of the past period. The length of growing seasons has less variable onset and cessation dates, while in the future, the length of the growing period (LGP) of 174 to 177 days will be suitable for short- and long-cycle crops, as well as double cropping, benefiting crop production yield in the North-Western Ethiopian highlands in the future.

**Keywords:** rainfall; temperature; potential evapotranspiration; soil water content; climate projection





## 1. Introduction

There is a consensus that the impacts of climate change on agriculture will significantly contribute to the development challenges of ensuring food security and reducing poverty, particularly in Africa. Climate change plays a crucial role in agricultural production, directly impacting processes from the start of land preparation to the final harvest [1–4]. In the past half-century, Ethiopia has experienced a late onset and early cessation of seasonal rains, leading to the associated failure of the crop growing season [5,6]. To optimally utilize seasonal rainfall for agricultural production, additional knowledge and understanding are needed regarding how shifts in rainfall seasons may affect crop yields. Addressing this issue requires tailored predictive information on the start and end of the growing season, which is timely and important and can assist in adopting appropriate adaptation and coping measures.

Climate change has various impacts on the growth and development processes of crops. For instance, an increase in carbon dioxide ($CO_2$) can stimulate photosynthesis rates, sometimes resulting in higher yields [7,8]. Changes in temperature and precipitation affect crop photosynthesis, plant development rates, as well as water and nutrient budgets in the field [9,10]. One of the critical processes affected by climate change is photosynthesis. The direct effects of $CO_2$ enrichment on plants include an increase in the rate of photosynthesis and water-use efficiency (WUE). As $CO_2$ concentrations increase, the transpiration intensity of plants reduces by partially closing the stomata, leading to improved water use efficiency and a lower probability of water stress occurrence. These physiological responses are known as the $CO_2$ fertilization effect or the direct effect of increased $CO_2$.

The increase in temperature due to climate change has both positive and negative impacts on crop production. For example, in the middle and higher latitudes and at high elevations in the tropics, global warming will extend the potential growing season's length, allowing for earlier planting, maturation, and harvesting. Increased $CO_2$ may enable the completion of two or more cropping cycles during the same season and the expansion of crop-producing areas to higher latitudes and elevations, provided soil fertility is maintained [7,11,12]. In warmer, lower latitude regions, increased temperatures may accelerate the release of $CO_2$ through plant respiration, creating less-than-optimal conditions for net growth. High temperatures reduce yield by hastening maturation, not allowing crops to progress slowly through the season for maximum resource capture and assimilated partitioning to reproductive structures [9,13].

Therefore, under conditions that are physiologically 'too warm' for certain crops, yields may decrease. An increase in average temperature can have several effects, including lengthening the growing season in regions with a relatively cool spring and fall, and adversely affecting crops in regions where summer heat already limits production. Soil evaporation rates may increase, and the chances of severe droughts may also rise [14,15]. Warmer temperatures in high-altitude areas may allow more insects to overwinter in these areas. Crop damage from plant diseases is likely to increase in temperate regions because many fungal and bacterial diseases have a greater potential to reach severe levels when temperatures are warmer, or precipitation increases [16].

Changes in rainfall can affect soil erosion rates and soil moisture, both of which are crucial for growth and yields. In the case of the NW Ethiopian highlands, the impact of climate change is likely to be both positive and negative due to the area's different topography with high-, middle-, and low-altitude elevations.

Moreover, improved forecasts of seasonal precipitation with adequate spatial resolution could potentially increase agricultural production and reduce production risks [17]. Therefore, predictions of local determinants of climate change, climate variability, and the growing season are crucial for livelihoods dependent on rain-fed agriculture. Hence, this study uses fine-resolution climate change models to simulate and project climatic conditions and associated growing seasons in the NW Ethiopian highlands.

According to the World Bank's [18] categorization, the NW Ethiopian highlands consist of majority areas with substantially higher rainfall and more availability of arable land, where the average crop yield is double that of the whole country. Despite being potential zones for crops and livestock, the NW Ethiopian highlands experience erratic rainfall in terms of both spatial and temporal distribution, with dry spells significantly reducing crop yields and sometimes leading to total crop failure. This has caused the highlands to face recurrent drought [18].

The Ethiopian highlands, major contributors to crop production for the country, were affected by inter-annual and seasonal variability of rainfall, causing fluctuations in cereal production in the region [19,20]. Hence, research on climate change's impact on crop production and agricultural water management is timely and important to enhance food security in the country. Although some studies have been conducted to identify the spatial and temporal climate changes of the past and future, these have not specifically been tied to the growing season at the local level [3,4,21,22].

Moreover, the impact analysis of climate change on soil water content, and the onset and cessation date and length of the growing period derived from soil water content, provide more reliable results than other methods used to determine crop growing seasons. Hence, this research analyzes climate change and its impact on the length of the growing season (based on soil water content) in the Ethiopian highlands. The research results can be used to design suitable adaptation and mitigation strategies, as well as coping mechanisms. Thus, in this study, we address the past (1981–2010) and future (2041–2070) CMIP5 RCP6 model projections of trends and changes in specific climatic parameters (rainfall, maximum and minimum temperatures, PET, and soil water content) and their impact on the length of crop growing seasons in the NW Ethiopian highlands.

## 2. Materials and Method

### 2.1. Study Area

Figure 1 shows the location of NW Ethiopia and the distribution of the 35 meteorological stations utilized in the study. The NW Ethiopian highlands are situated in the north-western part of Ethiopia, located within 8–14° N and 35–39° E (Figure 1).

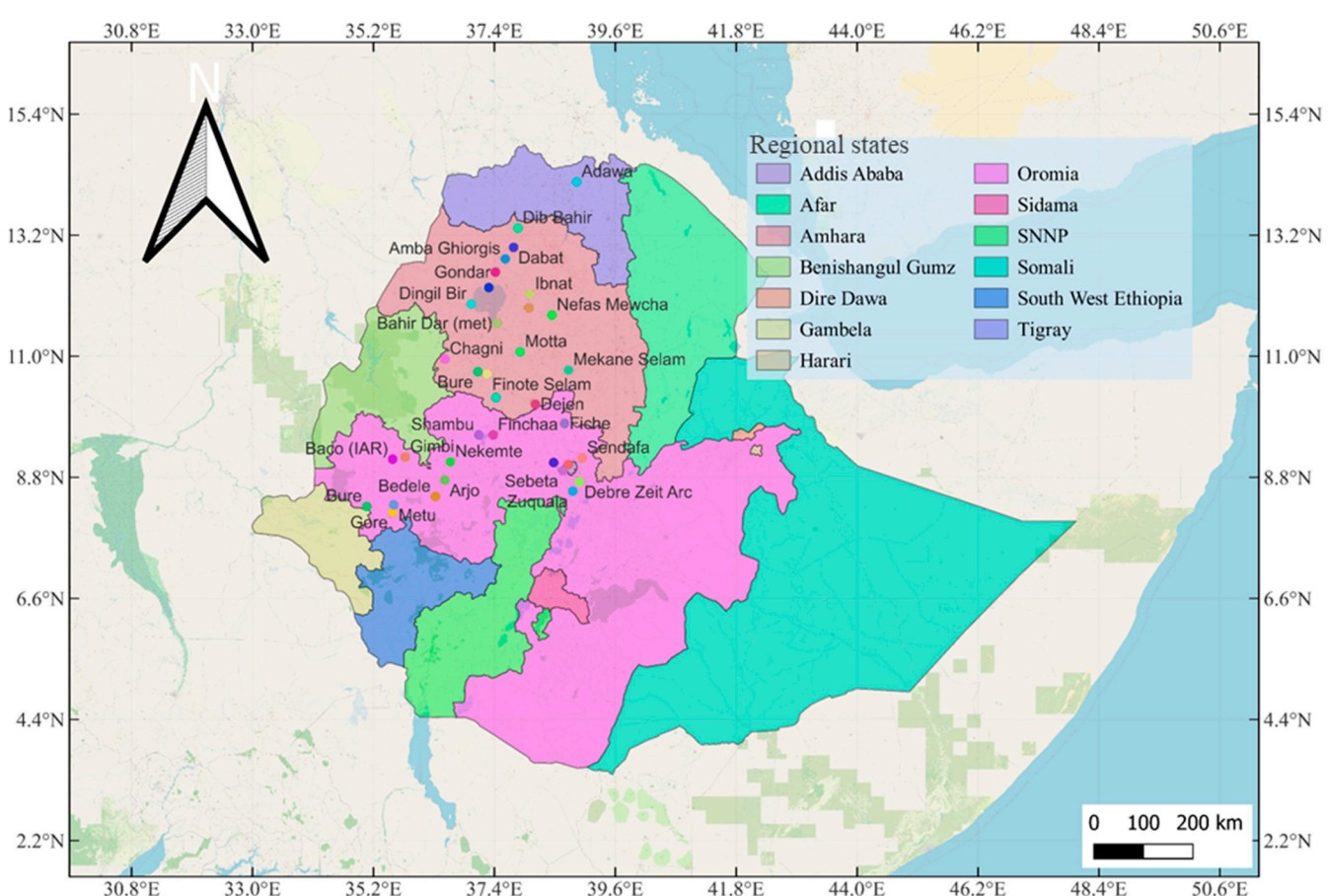

**Figure 1.** Location of NW Ethiopian highlands and distribution of the 35 meteorological stations utilized in this study.

Agriculture of the Study Area

Agriculture has been practiced for centuries in Ethiopia, with the overall farming system strongly oriented towards grain production as a source of livelihood and a way of life. Ethiopian peasant agriculture is characterized by polyculture, involving the traditional rotation of crops and livestock. This agricultural approach has persisted for centuries and continues to be beneficial for both the land and the people, meeting complete livelihood and nutritional needs, as well as ensuring community security. Despite remaining subsistent,

peasant agriculture allows individuals to produce everything required for themselves and their families, with a variety of crops being cultivated.

The proximity of areas with different altitudes has contributed to the diversity of crops grown. Farmers primarily focus on food crops, with only a few cash crops intended for the local market or barter system. However, Ethiopia has experienced droughts in the recent half-century [6,23], mainly due to the late onset and early cessation of seasonal rains, leading to the failure of the crop growing season. This is particularly challenging as the agricultural practices of most farmers in the country rely on rain-fed agriculture. To optimally utilize seasonal rainfall for agricultural production, additional knowledge is needed on how shifts in rainfall seasons may affect crop yields.

On the other hand, the soil type in the Ethiopian highlands is predominantly Vertisols, accounting for 12.7 million hectares in Ethiopia, with 7.6 million hectares in the NW Ethiopian highlands. Waterlogging of Vertisols is more severe in the Ethiopian highlands, where rainfall is higher and evaporative demands are lower. The technical efficiency of traditionally applied surface drainage techniques is not sufficient to fully utilize the potential of these soils.

*2.2. Data Sets and Research Design*

The data sources utilized for analysis included gauge data from weather stations, gridded observations (reanalysis) data, and CMIP5 rcp6 (representative concentration path 6) model simulation datasets extracted from the KNMI-Climate Explorer website (https://climexp.knmi.nl/start.cgi) (accessed on 23 May 2023). The Royal Netherlands Meteorological Institute (KNMI) hosts the Climate Explorer website, providing access to CMIP5 data. KNMI collaborates closely with international meteorological networks, such as the European Centre for Medium-Range Weather Forecasts (ECMWF) and the European meteorological satellite network EUMETSAT. The KNMI Climate Explorer (CE) web page, organized by [24], is a valuable resource. For the list of CMIP5 model groups open to the public on KNMI CE, see Supplementary Information (SI) Table S1.

This study focused on using CMIP5 rcp6 model data sources developed by the AIM modeling team at the National Institute for Environmental Studies (NIES) in Japan. The decision to use only CMIP5 rcp6 model groups is explained as follows. (1) CMIP5 rcp6 is a stabilization scenario where total radiative forcing stabilizes shortly after 2100, without over-shot, through the application of various technologies and strategies to reduce greenhouse gas emissions [25,26]. This aligns with the global consensus on reducing carbon emissions and mitigating climate change. (2) CMIP5 rcp6 models consist of 23 to 25 models serving as data sources for 21 climate variables. All these models' outputs were considered with equal weight given to each model or version with different parameterizations. Describing each CMIP5 rcp6 model is beyond the scope of this study; readers are referred to [27] and the Program for Climate Model Diagnosis & Intercomparison (https://shorturl.at/zHLPV) (accessed on 10 April 2023). (3) Furthermore, studies conducted in the Ethiopian highlands by [28,29] confirm minimal bias using CMIP5 rcp6 models. The average CMIP5 rcp6 model data annual cycle correlation results with GPCC V6 rainfall (r ≥ 0.6) and with CRU TS v3.22 maximum temperature (r ≥ 0.6) in the Ethiopian highlands indicate the skill of rcp6 in capturing rainfall and temperature (Figure 2a,b). Due to the above justifications and limited capacity, only CMIP5 rcp6 model data sources were used to project the climatic conditions of the NW Ethiopian highlands. It should be noted that the results shown in this study are from the CMIP5 rcp6 scenario only. Nevertheless, when assessing impacts, changes in climate should be examined in the remaining three scenarios as well.

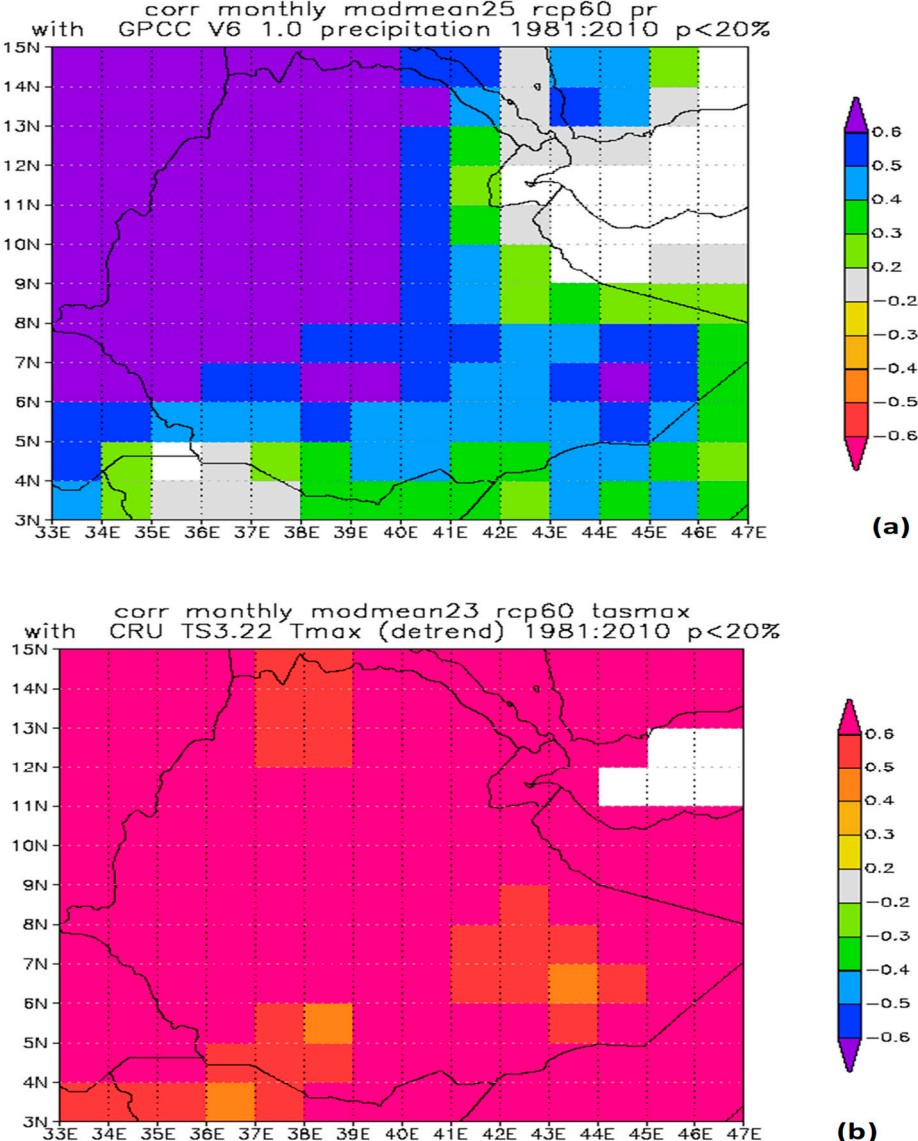

**Figure 2.** Average CMIP5 rcp6 correlation (**a**) with GPCC V6 for rainfall and (**b**) with CRU TS v3.22 for tasmax.

Each of the variables—rainfall, sensible heat flux (SHF), and temperature—was area-averaged for the NW Ethiopian highlands to facilitate temporal and spatial analysis. The procedures and methods devised to calculate soil water content, incorporating Runoff (RO) (not included in the Thornthwaite soil water balance method) and utilizing the Thornthwaite and Mather soil moisture retention table, play a significant role in studying soil moisture in the NW Ethiopian highlands [30]. The soil moisture changes obtained from the computed soil water content were applied to determine the onset date, cessation date, and length of growing period (LGP) for the study area.

### 2.3. Determining the Best Reference for Validation

Before assessing the models, the optimal reference values for temperature, rainfall, and potential evapotranspiration (PET) variables were determined. For tasmax, the Climate Research Unit Time Series (CRU TS v3.22) interpolated observations compared with the average of 35 stations over the study area from 1981 to 2000. The correlation was found to be r = 0.95, with a minor cold bias of $-0.09$ °C. This suggests that CRU TS v3.22 is a reliable data source for temperature, suitable for validating and evaluating the CMIP5 rcp6 models' temperature data sources, and for selecting the most appropriate model.

Comparing Global Precipitation Climatology Center version 6 (GPCC v6) rainfall with the area-average of 35 station observations from 1981 to 2000, a correlation of r = 0.99 was observed, with a minimal dry bias of GPCC −0.6 mm/day based on the scatter plot. Therefore, GPCC v6 at 50 km resolution effectively accounts for Ethiopia's orographic effect on rainfall. It can be used for validating and evaluating the CMIP5 rcp6 models' data sources for precipitation and selecting the most suitable model. The selected climate model for rainfall simulation was then applied to derive the soil water content and the length of the growing season in the NW Ethiopian highlands for both the past (1981–2010) and the future (2041–2070). In line with this, GPCC rainfall products have demonstrated good agreement with reference gauge data in studies conducted by [31,32] on the Ethiopian highlands.

PET was calculated using 18 stations from [33,34] in the NW Ethiopian highlands, with Climate Research Unit–Potential Evapotranspiration (CRU-PET) used as a reference for comparison with other datasets and proxies, such as maximum temperature and surface heat flux. The relation between station PET and CRU maximum temperature was significant but considering the known lag of temperature behind solar radiation (Ojo, 1977), it was excluded. In comparison with ECMWF reanalysis-adjusted sensible heat flux (adjshf), a robust relation was indicated by a regression of R2 = 0.95. Weaker correlations were found between station PET and Modern-Era Retrospective Analysis (MERRA) reanalysis-adjusted sensible heat flux. Thus, the European Centre for Medium-Range Weather Forecasts (ECMWF) ERA-Interim adjshf served as the validation reference for CMIP5 rcp6-sensible heat flux, following adjustment to match ECMWF.

### 2.4. Model Validation and Evaluation

The evaluation involved comparing monthly gridded observations, reanalysis-gridded data from GPCC v6 (for rainfall), CRU TS v3.22 (tasmax), and ERA-Interim (adjshf) with station observation data. The comparison revealed a correlation coefficient (r) greater than 0.90, which was used to validate the CMIP5 rcp6 model's area-averaged data over the NW Ethiopian highlands. Stepwise criteria were applied for comparison, and the models that met most criteria were CCSM4 (for tasmax) and HadGEM2-ES (for precipitation and PET/adjshf). Consequently, these two CMIP5 models were utilized for projecting climate change and calculating the soil water content. Additionally, they were employed to determine the length of the growing season for both the past and future in the study area. Following the stepwise criteria mentioned above, the evaluation and justification of CMIP5 rcp6 model simulations against GPCC V6 (rainfall), CRU TS3.22 (tasmax)-gridded observations, and ERA-Interim (shf) reanalysis-gridded observations were proposed in the context of the Ethiopian highlands' climate. This approach is consistent with the work of [29,32]. Details on how the reference models were selected for precipitation, temperature, and sensible heat flux are presented in Table 1, Table 2, and Table 3, respectively.

**Table 1.** NW Ethiopian highlands CMIP5 rcp6 model's rainfall stepwise evaluation taking GPCC v6 as reference from 1981–2010. CMIP5 rcp6 rainfall models validation taking GPCC V6 as reference from 1981–2010.

| No | Model | GPCC v6(r) | JJAS- Diff. (GPCC v6 vs. rcp6) | Season Peak-JA/Day) | Annual Pattern Fitness | Remark |
|----|-------|-----------|-------------------------------|---------------------|------------------------|--------|
| 1 | bcc-csm1-1 | 0.65 | −3.2 | 3.8 | poor | Screened out |
| 2 | bcc-csm1-1-m | 0.90 | −1.4 | 6.3 | poor | Screened out |
| 3 | CCSM4 | 0.76 | −2.0 | 4.3 | poor | Screened out |
| 4 | CESM1-CAM5 | 0.75 | −0.9 | 5.7 | poor | Screened out |
| 5 | CSIRO-Mk3-6-0 | 0.91 | −1.1 | 8.8 | moderate | Screened out |
| 6 | FIO-ESM | 0.80 | −2.1 | 4.6 | poor | Screened out |
| 7 | GFDL-CM3 | 0.87 | −1.5 | 6.2 | poor | Screened out |

**Table 1.** *Cont.*

| No | Model | GPCC v6(r) | JJAS- Diff. (GPCC v6 vs. rcp6) | Season Peak-JA/Day) | Annual Pattern Fitness | Remark |
|---|---|---|---|---|---|---|
| 8 | GFDL-ESM2G | 0.86 | −0.3 | 6.7 | moderate | Screened out |
| 9 | GFDL-ESM2M | 0.87 | −0.5 | 6.4 | moderate | Screened out |
| 10 | GISS-E2-H_p1 | 0.96 | −5.3 | 3.2 | poor | Screened out |
| 11 | GISS-E2-H_p2 | 0.96 | −5.5 | 2.8 | poor | Screened out |
| 12 | GISS-E2-H_p3 | 0.96 | −4.7 | 4.0 | poor | Screened out |
| 13 | GISS-E2-R_p1 | 0.97 | −5.6 | 2.5 | poor | Screened out |
| 14 | GISS-E2-R_p2 | 0.97 | −5.7 | 2.6 | poor | Screened out |
| 15 | GISS-E2-R_p3 | 0.96 | −5.3 | 2.7 | poor | Screened out |
| 16 | HadGEM2-AO | 0.96 | −0.4 | 8.1 | high | 2nd selected |
| 17 | HadGEM2-ES | 0.97 | −0.5 | 8.0 | high | 1st selected |
| 18 | IPSL-CM5A-LR | 0.89 | −2.7 | 6.3 | poor | Screened out |
| 19 | IPSL-CM5A-MR | 0.90 | −3.3 | 5.3 | poor | Screened out |
| 20 | MIROC5 | 0.98 | 11.5 | 21.1 | poor | Screened out |
| 21 | MIROC-ESM | 0.86 | −0.5 | 7.3 | high | Screened out |
| 22 | MIROC-ESM-CHEM | 0.88 | −0.5 | 7.4 | high | Screened out |
| 23 | MRI-CGCM3 | 0.93 | −2.5 | 6.0 | moderate | Screened out |
| 24 | NorESM1-M | 0.66 | −2.7 | 3.4 | poor | Screened out |
| 25 | NorESM1-ME | 0.62 | −2.3 | 3.3 | poor | Screened out |

**Table 2.** CMIP5 rcp6 models' maximum temperature (tasmax) stepwise evaluation taking CRU TS3.22 as reference from 1981–2010.

| CMIP5 rcp6 Tasmax Model Stepwise Evaluation Using CRUTS 3.22 as Reference from 1981–2010 | | | | | |
|---|---|---|---|---|---|
| No | Model Type | CRUTS 3.22 Tasmax (r) | Seasonal Peak/MAM Diff. (CRUTS 3.22 vs. rcp6) | Annual Pattern Fitness | Remark |
| 1 | bcc-csm1-1 | 0.90 | 2.0 | Poor | Screened out |
| 2 | bcc-csm1-1-m | 0.35 | −1.4 | Poor | Screened out |
| 3 | CCSM4 | 0.98 | −0.5 | High | 1st selected |
| 4 | CESM1-CAM5 | 0.90 | −2.3 | Poor | Screened out |
| 5 | CSIRO-Mk3-6-0 | 0.43 | −1.8 | Poor | Screened out |
| 6 | FIO-ESM | 0.81 | 1.6 | Poor | Screened out |
| 7 | GFDL-CM3 | 0.90 | 3.6 | Poor | Screened out |
| 8 | GFDL-ESM2G | 0.91 | −1.0 | Moderate | Screened out |
| 9 | GFDL-ESM2M | 0.92 | −0.7 | Moderate | 4th selected |
| 10 | GISS-E2-H_p1 | 0.94 | 4.0 | Poor | Screened out |
| 11 | GISS-E2-H_p2 | 0.90 | 4.0 | Poor | Screened out |
| 12 | GISS-E2-H_p3 | 0.95 | 3.9 | Poor | Screened out |
| 13 | GISS-E2-R_p1 | 0.97 | 4.7 | Poor | Screened out |
| 14 | GISS-E2-R_p2 | 0.93 | 4.8 | Poor | Screened out |
| 15 | GISS-E2-R_p3 | 0.95 | 4.8 | Poor | Screened out |
| 16 | HadGEM2-AO | 0.91 | 0.2 | Moderate | 3rd selected |

**Table 2.** *Cont.*

| | CMIP5 rcp6 Tasmax Model Stepwise Evaluation Using CRUTS 3.22 as Reference from 1981–2010 | | | | |
|---|---|---|---|---|---|
| No | Model Type | CRUTS 3.22 Tasmax (r) | Seasonal Peak/MAM Diff. (CRUTS 3.22 vs. rcp6) | Annual Pattern Fitness | Remark |
| 17 | IPSL-CM5A-LR | 0.20 | −1.1 | Poor | Screened out |
| 18 | IPSL-CM5A-MR | 0.22 | 1.1 | Poor | Screened out |
| 19 | MIROC5 | 0.93 | −3.3 | Poor | Screened out |
| 20 | MIROC-ESM | 0.93 | 3.2 | Poor | Screened out |
| 21 | MIROC-ESM-CHEM | 0.94 | 2.8 | Poor | Screened out |
| 22 | MRI-CGCM3 | 0.85 | 0.9 | Poor | Screened out |
| 23 | NorESM1-M | 0.94 | 0.3 | High | 2nd selected |

**Table 3.** Stepwise evaluation results comparing CMIP5 rcp6 model sensible heat flux (SHF) with adjshf ECMWF-ERA-int. to simulate adjshf for the NW Ethiopian highlands based on the period 1981–2010.

| No | CMIP5 rcp6 Models | ERA Vs rcp6 PET/Adjshf (r) | Seasonal Diff. FMA (ERA vs. rcp6) | Annual Cycle Fitness | Remark |
|---|---|---|---|---|---|
| 1 | bcc-csm1-1-m | 0.79 | 0.46 | moderate | Filtered out |
| 2 | bcc-csm1-1 | 0.73 | 0.72 | poor | Filtered out |
| 3 | CCSM4 | 0.96 | 0.63 | high | 2nd selected |
| 4 | CESM1-CAM5 | 0.78 | 0.06 | moderate | Filtered out |
| 5 | CSIRO-Mk3-6-0 | 0.79 | 1.04 | poor | Filtered out |
| 6 | FIO-ESM | 0.95 | 0.81 | high | 4th selected |
| 7 | GFDL-CM3 | 0.92 | 1.26 | poor | Filtered out |
| 8 | GFDL-ESM2G | 0.9 | 0.92 | moderate | Filtered out |
| 9 | GFDL-ESM2M | 0.92 | 0.88 | moderate | Filtered out |
| 10 | GISS-E2-H_p1 | 0.95 | 1.48 | poor | Filtered out |
| 11 | GISS-E2-H_p2 | 0.96 | 1.32 | poor | Filtered out |
| 12 | GISS-E2-H_p3 | 0.97 | 1.21 | poor | Filtered out |
| 13 | GISS-E2-R_p1 | 0.95 | 1.29 | poor | Filtered out |
| 14 | GISS-E2-R_p2 | 0.97 | 1.11 | poor | Filtered out |
| 15 | GISS-E2-R_p3 | 0.97 | 1.04 | moderate | Filtered out |
| 16 | HadGEM2-AO | 0.93 | 0.34 | high | Filtered out |
| 17 | HadGEM2-ES | 0.96 | 0.24 | very high | 1st selected |
| 18 | IPSL-CM5A-LR | 0.87 | 1.52 | poor | Filtered out |
| 19 | IPSL-CM5A-MR | 0.74 | 1.34 | poor | Filtered out |
| 20 | MIROC5 | 0.95 | −0.95 | moderate | Filtered out |
| 21 | MIROC-ESM | 0.87 | 0.11 | moderate | Filtered out |
| 22 | MIROC-ESM-CHEM | 0.85 | 0.05 | moderate | Filtered out |
| 23 | MRI-CGCM3 | 0.8 | 0.31 | moderate | Filtered out |
| 24 | NorESM1-M | 0.95 | 0.58 | high | 3rd selected |
| 25 | NorESM1-ME | 0.91 | 0.44 | high | Filtered out |

*2.5. Climate Change Analysis*

Among the various approaches to comprehend climate change, the technique of subtracting past data from future projections is frequently employed. While this method is straightforward, it is plagued by uncertainties stemming from natural climate variability, contingent upon the chosen variable and time periods. In this study, the climatological reference period of 30 years [35] from 1981 to 2010 is utilized. This timeframe strikes a balance between the extended duration necessary for precipitation assessments and a shorter one sufficient for temperature analyses. Another consideration is that trends observed in the most recent 30-year period hold greater relevance for predicting the subsequent 30-year period, particularly considering the acknowledged acceleration of greenhouse warming. To evaluate the signal-to-noise ratio, linear trends are employed instead of comparing differences between two periods. These trends can be expressed as proportional to $CO_2$ concentration or global mean temperature. Despite the accelerating trends in $CO_2$ concentration (Figure 2), the temporal response of climate is assumed to be linear during the study period. This assumption helps eliminate ambiguities arising from second-order responses. Examining trends in rainfall and temperature can reveal 'drift' attributed to climate change, whether natural or anthropogenic. Detecting such trends in the data is valuable for decision making at the policy level in agriculture and food security. Trend analysis for temperature, rainfall, PET, and anomalies involved fitting a linear equation through regression. The InStat + v3.36 software [36] applied *t*-tests to determine the statistical significance of the changes. Bias was assessed by comparing the simulated or projected annual cycle against reference data for all the variables mentioned. Negative bias occurs when the model minus reference is less than 0, representing dry bias for rainfall and cold bias for temperature. Statistical measures of association, indicating the strength or degree of the relationship between variables, were determined spatially and temporally. The Royal Netherlands Meteorological Institute-Climate Explorer (KNMI CE) climexp.knmi.nl webpage and Excel 2007, along with InStat +v3.36 software, were used for this purpose. The 95% significance level served as the threshold for validity.

*2.6. Potential Evapotranspiration*

Among the various methods employed for calculating potential evapotranspiration (PET), the FAO Penman–Monteith method is recommended by the FAO as the standard [37]. However, in developing countries such as Ethiopia, obtaining the necessary climate parameters for calculating PET is challenging. Due to this difficulty, an alternative method was employed, involving the comparison of other climate variables expected to serve as substitutes for PET. To implement this surrogate method, PET values calculated from station observations by [33,34] in the NW Ethiopian highlands, along with PET estimates from the CRU data source, were used as references. These values were then compared with CRU (tasmax), ECMWF, and MERRA reanalysis data for sensible heat flux (shf), latent heat flux (lhf), and the sum of shf and lhf. The comparison results, which showed good agreement with PET station observations and CRU-PET estimates, were used to produce a surrogate PET. Subsequently, this surrogate PET was utilized to justify the data from CMIP5 rcp6 models for the purpose of evaluation and analysis.

*2.7. Soil Water Content Proxy*

The spatial distribution and temporal variation of soil water content in the top meter of the Earth's surface are crucial for numerous environmental studies. Soil water content can be determined through (i) point measurements; (ii) soil water content models; and (iii) remote sensing. In this study, the soil water content calculated from CMIP5 rcp6 models' data was utilized to estimate soil water balance and changes in soil water content for both the past and the future. Subsequently, using precipitation and a surrogate for PET from CMIP5 rcp6 models, the soil water balance estimate (P-Es-RO) was calculated for both

historical and future periods. These estimates were employed for the analysis of soil water balance in the region as shown in Equation (1). RO (runoff) will be calculated as follows:

$$RO = Cro \times P \tag{1}$$

where the runoff coefficient (Cor) calculate by estimating Cro for each site, taking into account the soil type, steepness, and the slope range, as per [38], Table S1.

*2.8. Growing Season Analysis*

The LGP methodology draws on insights from [39–42]. The initiation of the growing season necessitates an increase in soil water content, contingent on both soil and crop type. When the change in soil water first becomes positive ($\Delta ST > 0$), the season is considered to have commenced with germination. Conversely, the cessation time is determined when the change in soil water content becomes zero or negative ($\Delta ST \leq 0$). This methodology is applied consistently from past simulations to future projections. Despite Ethiopia being a tropical country, the highlands experience minimum temperatures below 9 °C, a critical threshold beyond which crops can no longer extract available soil water [39–42].

Upon identifying the CMIP5 model closest to the observations, the study area's growing season length is estimated using a proxy of water available for storage (P-Es-RO) and RO calculated from (P) and the runoff coefficient (Cor) Equation (1). The calculation considers Cro between 0.15 and 0.32, representing heavy soil, with slope ranging from 2% to 7%, following [29] for details. The computation continues until the last month of the wet season, when water available for storage equals or exceeds the soil's water-holding capacity. The length of the growing season is then estimated from the CMIP5 rcp6.0 projection of soil water change ($\Delta ST$) obtained through the calculation of (P-Es-RO), as outlined below, drawing on [40,43–45].

The initial step involves determining the soil, vegetation, crop type, root depth, and water-holding capacity of the study area to obtain corresponding values from the soil moisture retention tables of [30]. Given that the major crops in the NW Ethiopian highlands are corn and cereal grains, the area is classified as moderately deep-rooted crops with soil type silt loam, water holding capacity 200 mm/m, root zone 100 cm, and an applicable soil moisture retention table of 200 mm, used to estimate soil water content.

Accumulated potential water loss represents the potential deficiency of soil moisture associated with moisture contents below the water-holding capacity of the soil. This accumulation increases during dry seasons, decreases during wet seasons due to soil moisture recharge, and equals zero when the soil moisture storage matches the water-holding capacity. The next step involves computing the accumulated potential water loss (*Acc. Pot. WL*), starting from an estimated value of potential water deficiency in the first month after the rainy season when ($P - Es - Ro$) turns negative (as per [30]). Subsequent monthly values are obtained by progressively adding ($P - Es - Ro$) with negative values (Equation (2)).

$$Acc.Pot.WL_{i+1} = Acc.Pot.WL_{i+1} + \left( P - Es - Ro \right)_{i+1} \tag{2}$$

where $i$ is the current month and $i + 1$ the next month from the current month. If the soil moisture exceeds the water-holding capacity, the soil moisture is set equal to the water-holding capacity:

$$S_{m(j)} = \begin{cases} S_{m(j-1)} + (P - Es - Ro)_{(j)} & for \; S_{m(j-1)} + (P - Es - Ro)_{(j)} < W_{WHC} \\ W_{WHC} & for \; S_{m(j-1)} + (P - Es - Ro)_{(j)} W \geq_{WHC} \end{cases} \tag{3}$$

where $S_{m(j)}$ is the soil moisture storage (in millimeters) for the $j$-th month and $W_{WHC}$ is the water-holding capacity of the soil.

The moisture retained in the soil is determined by converting each value of accumulated potential water loss into soil water content storage (ST) with the help of a soil moisture retention table. The change in soil moisture storage is calculated as the difference between the current and the previous month's soil moisture storage, starting from December of each year and moving into the next growing season. The change in soil moisture is considered positive if the soil moisture has increased and negative if it has decreased. Once ΔST is obtained, the onset date, cessation, and subsequent length of growing period (LGP) are estimated from monthly data. To better resolve the time scale, ΔST is interpolated to daily data using INSTAT v3.36 with a 10-day running mean. The daily onset date, cessation date, and LGP are then estimated. Finally, a minimum temperature threshold is applied, which may induce an early cessation of LGP. The flow chart illustrating how the research steps interlink can be found in Supplementary Information (SI), Figure S1, and a summary of the methods used in the research is presented in Table S2.

## 3. Results and Discussion

### 3.1. Changes in Temperature

In the area of study, according to the CCSM4 model projection, the average tasmax is expected to rise from 26.5 °C in the past (1981–2010) to 27.8 °C in the future (2041–2070), indicating a projected increase of 1.3 °C when comparing the past and future tasmax. For average tasmin, the model projection shows a change from 15.6 °C in the past to 16.8 °C in the future, with an expected rise of 1.2 °C in the future. The temperature trend test for both tasmax and tasmin across all time periods (past, future, and forward projection) revealed statistically significant and increasing trends.

The increasing trend for tasmax (Figure 3b) anticipates tasmax rises of about +0.023 °C/yr, consistent with [46,47] using the RCP6 scenario (+0.025 °C/yr). Similar results have been reported in IPCC assessments of tasmax over Ethiopian/NW Ethiopian highlands by [29,48]. The [49] projection of average annual temperature using RCP4.5 at the end of this century is +2.0 °C, almost in the middle of the projection difference (+2.8 °C) obtained from the RCP6 scenario used in this study. It should be noted that RCP4.5 (650 ppm $CO_2$ eq) and RCP6 (850 ppm $CO_2$ eq) are stabilization scenarios without an overshot pathway after 2100. The trend analysis for tasmin (Figure 3c) and the intra-seasonal trend analysis of the hot season (February to May) and the cool season (November to January) show significant changes with temperature increases of +0.0271 °C/yr and +0.0226 °C/yr, respectively. These indicate that the hot season will heat up faster, while the cold season and tasmin will experience slower temperature increases. The annual cycle comparison of past and future tasmax and tasmin shows a high correlation ($R^2 = 0.99$) with higher temperature changes in the future for all months (≈+1.3 °C). Regarding the hot and cool seasons, the hot season in the past (February-May) remains consistent in the future, with an average temperature rise from 29.6 °C to 31.1 °C (+1.5 °C). The cool season, from November to January, also remains consistent in the past and future, with an average temperature rise from 13 °C to 14.3 °C (+1.3 °C) in the future (Figure 3a).

The evaluation of spatial trend maps calculated for the time period 1981–2100 using the CCSM4 model projection indicates fast warming in the NW Ethiopian highlands, with faster warming in the central highlands (>+0.026 °C/yr) slowing down (+0.020 to +0.025 °C/yr) toward the edges of the highlands, as shown in Figure 4a. Furthermore, tasmin shows slow warming in the western part (<+0.021 °C/yr), while in the remaining part of the study area, there is modest warming (+0.021 to 0.024 °C/yr), and the fastest warming is projected to occur in the northern extreme part (+0.024 to +0.025 °C/yr), as depicted in Figure 4b. These trends align with [6,48] and may not be entirely harmful in the region [46]. Nevertheless, projections from [50] of mean annual temperature increase by the 2050s may result in teff, barley, and sorghum yields declining by 8–17%, and wheat and maize increasing by 1–3%.

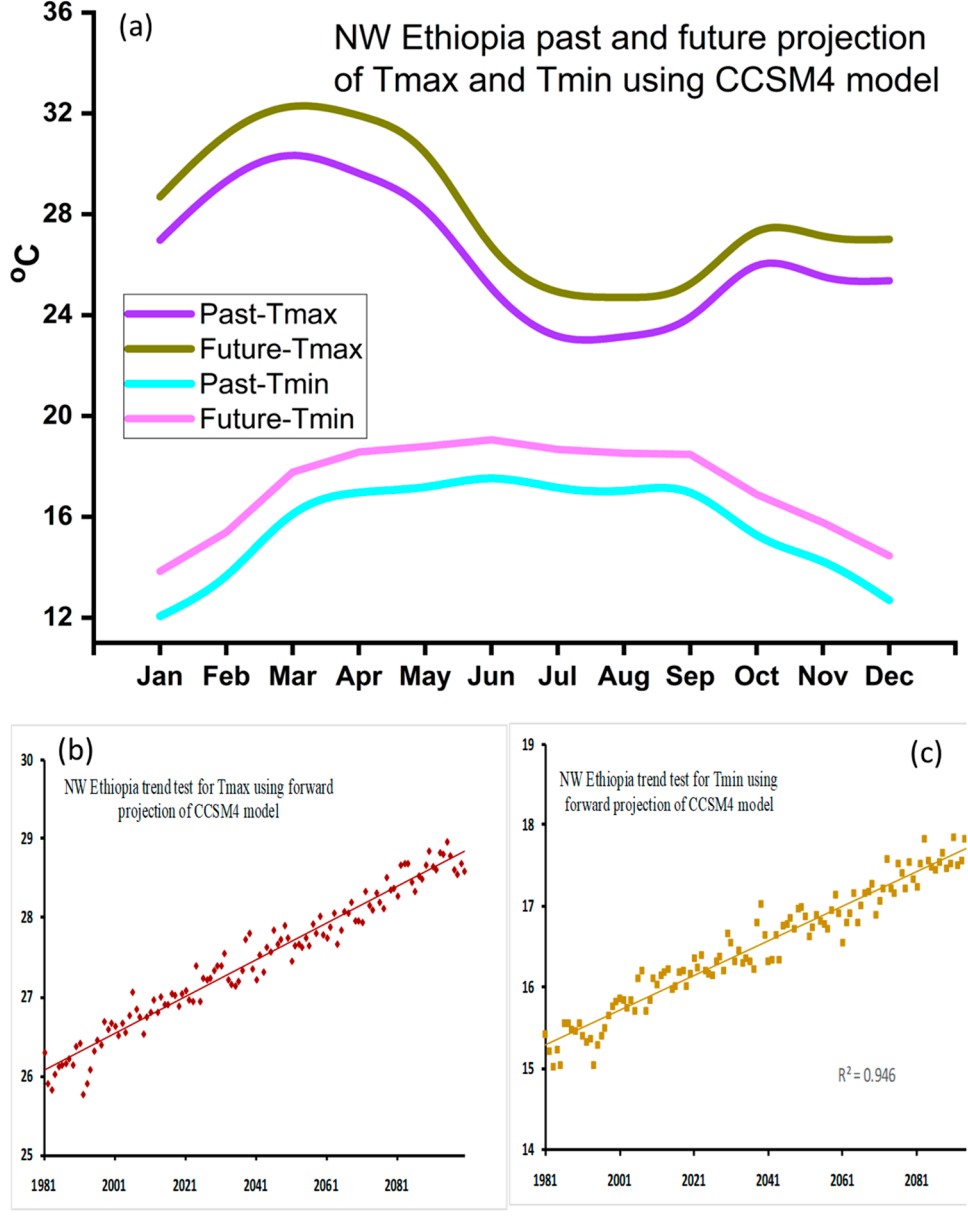

**Figure 3.** Annual cycle comparison of CCSM4 projection of past (1981–2010) (**a**) and the future (2041–2070) and regression trend test of CCSM4 forward projection (**b**) for tasmax and (**c**) for tasmin from 1981 to 2100.

The spatial distribution of tasmax (Figure 5a) depicts most of the NW Ethiopian highlands tasmax ranging between 23 and 26 °C, with the hottest part from 28 to 31 °C along the western border and the extreme north of the region, while the coolest (<22 °C) part is the south-central part of the study area. The spatial temperature change shows an increase in tasmax in the future by >+1.3 °C, but in the south-central part of the highlands, it will most likely increase faster by +1.45 °C (Figure 5b). The spatial pattern of mean surface air minimum temperature (tasmin) simulated by CCSM4 over Ethiopia clearly identifies cool highlands (<15 °C) surrounded by warmer lowlands (<19 °C). There are also areas in the southern central part that were the coolest (9 to 11 °C) (Figure 6a). The change between past and future CCSM4 minimum air temperatures over the NW Ethiopian highlands indicates warming (+1.13 to +1.37 °C), except for the central western part which warms slowly (+0.97 to +1.13 °C), while the northern extreme part warms faster (>+1.45 °C) (Figure 6b). The coolest zone (9 to 11 °C) in the past is anticipated to warm by +1.2 to 1.4 °C, implying the minimum temperature of this coolest zone will range between 10.2

and 12.4 °C in the future. The rising tasmin may improve the physiological development of crops, especially at the end of the growing season.

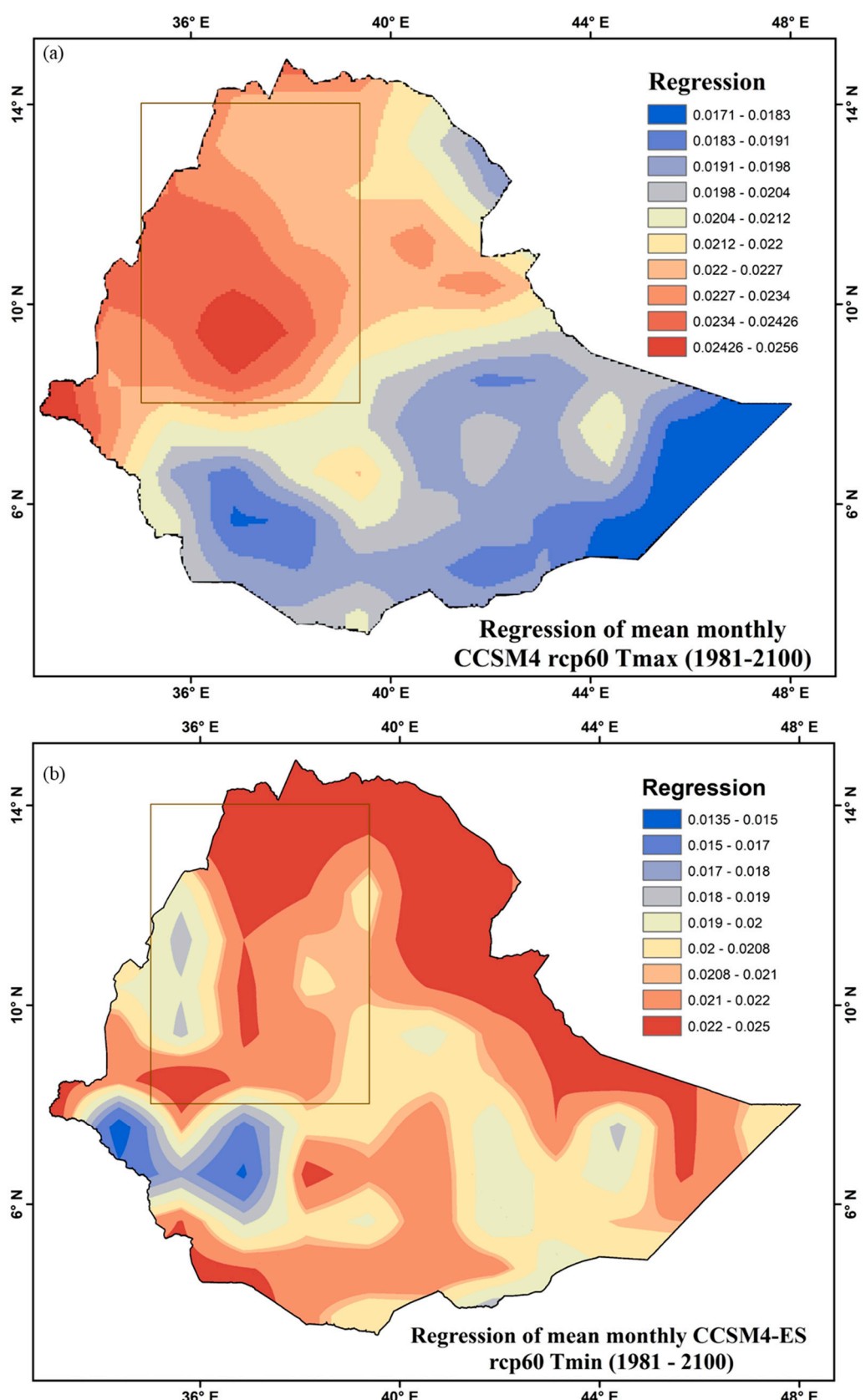

**Figure 4.** Spatial trend projection of CCSM4 for (**a**) tasmax and (**b**) tasmin from 1981 to 2100.

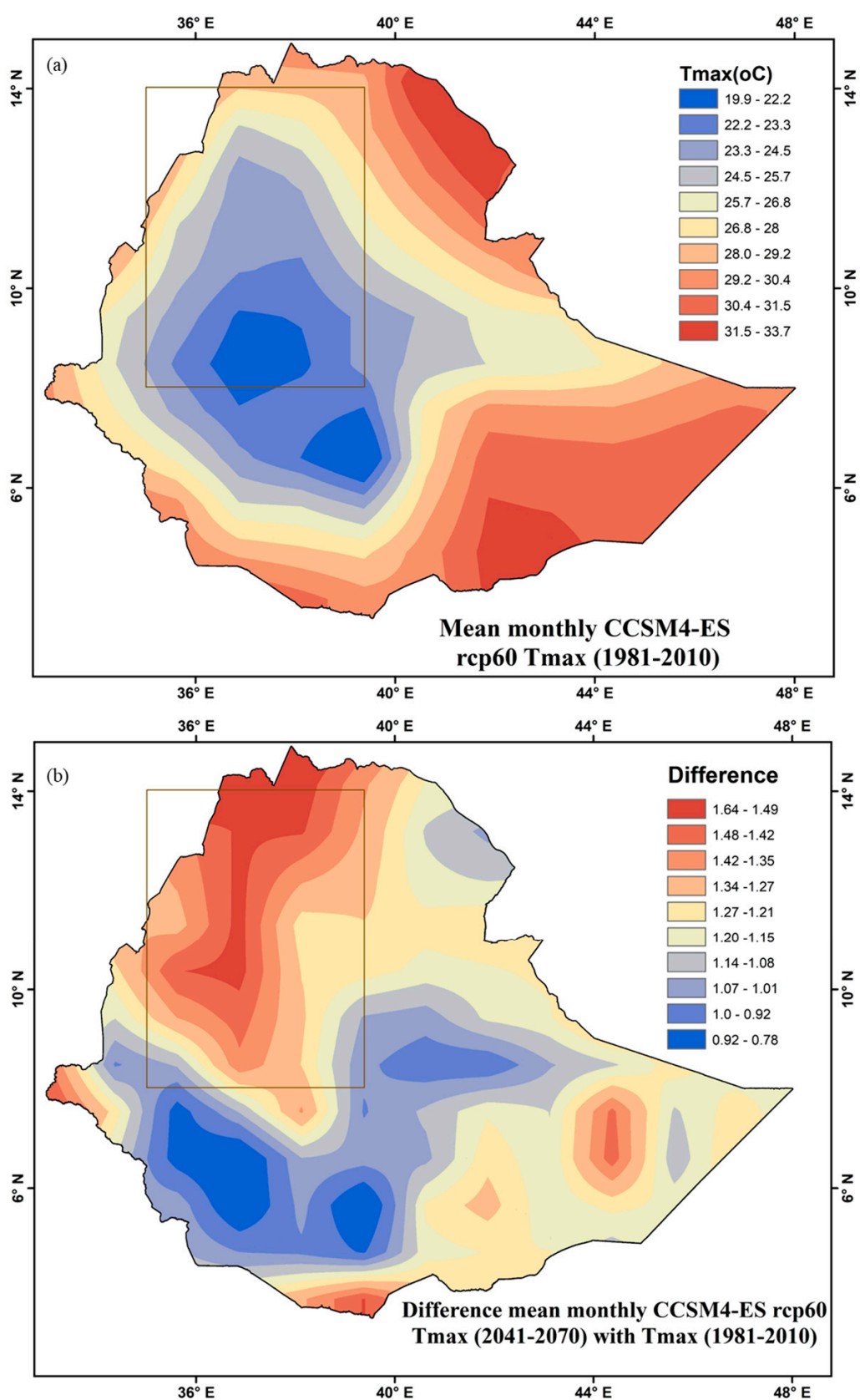

**Figure 5.** Average maximum temperature (Tasmax): (**a**) spatial distribution of the past and (**b**) spatial change in the future.

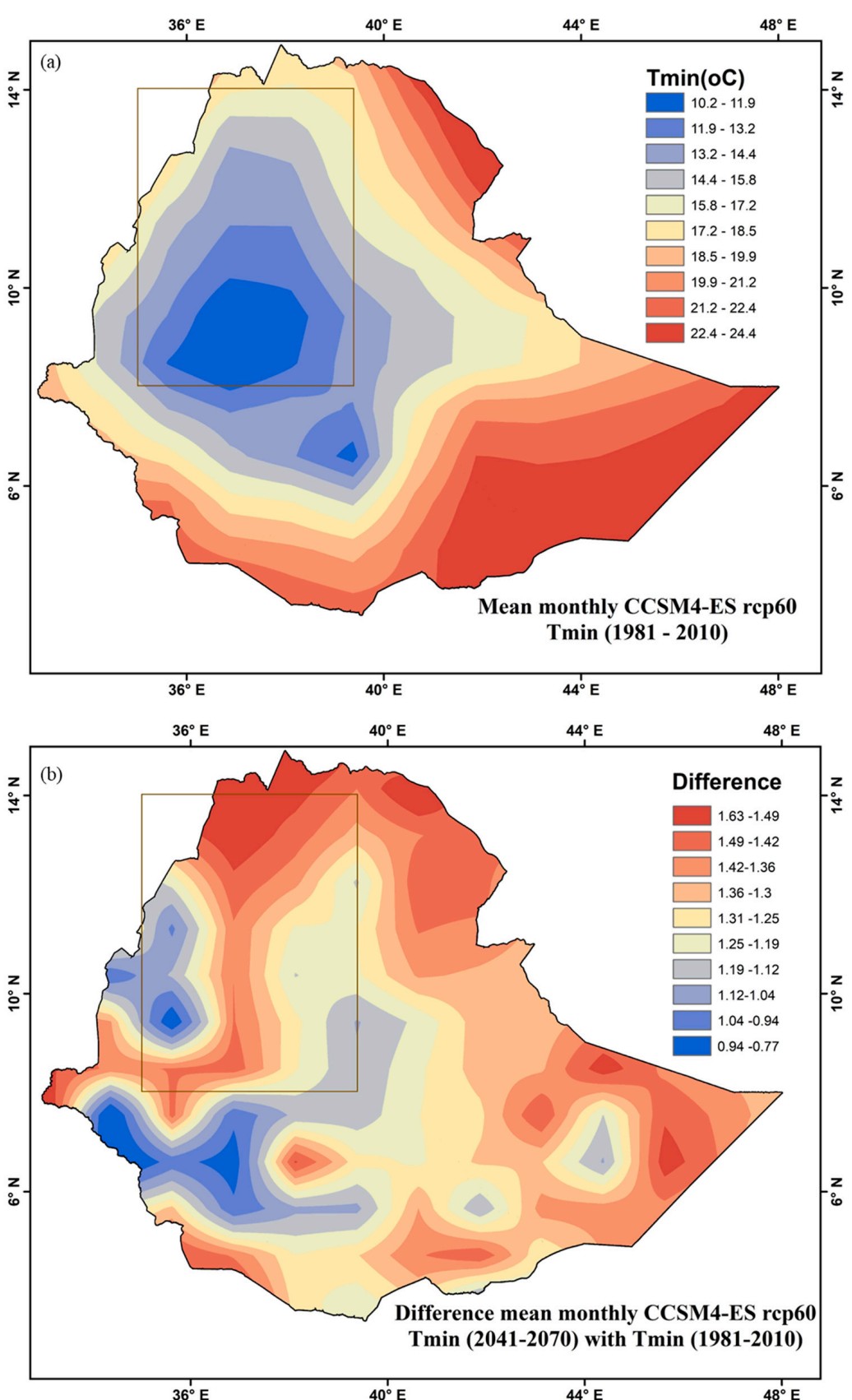

**Figure 6.** Average minimum temperature (tasmin): (**a**) spatial distribution of the past and (**b**) spatial change in the future.

There is high confidence in future warming, so there is a need for more information on the impacts of higher temperatures on evaporation (and soil water availability), agriculture, health, and vector-borne diseases [50,51]. The increase in temperature due to climate change has positive impacts on crop production at high elevations in the tropics by extending the length of the growing season, allowing earlier planting of crops, later maturation and harvesting, and even providing the opportunity for double cropping in a single season.

### 3.2. Changes in Rainfall

The results from the HadGEM2-ES simulation indicate that the mean average annual rainfall in the NW Ethiopian highlands was 1210 mm (1981–2010) and 1267 mm (2041–2070), reflecting an increase of +4.7% in the future. This finding aligns with [29,48–50,52,53]. The slope of the annual rainfall in the HadGEM2-ES forward projection (1981–2100) is upward by +0.0023 mm/day, demonstrating a statistically significant trend (Figure 7a), consistent with [29].

In the forward projection (1981–2100) of the seasons, the major rainfall season (JJAS) exhibits a trend of +0.0023 mm/day, and the dry season (ONDJ) shows a trend of +0.0006 mm/day, both indicating a significant increasing trend. The short rainfall season (FMAM) demonstrates a trend of +0.00031 mm/day but is not statistically significant. Anticipating wetter conditions, the livelihood of NW Ethiopian farmers may benefit from the increased rainfall, assisting in improving crop production (Figure 7a). The rainfall annual cycle of the past (1981–2010) and future (2041–2070) shows a unimodal nature, consistent with results reported by [5,54,55]. The main change in the future is a seasonal broadening with more rain in May-June and September-October (Figure 7b). The seasons' contribution to the annual rainfall during the forward projection (1981–2100) shows JJAS (67.1%), FMAM (22.6%), and ONDJ (10.3%), indicating that JJAS will be the major rainfall season, followed by FMAM, and ONDJ as the dry season of the study area. Commonly, JJAS and FMAM seasons are merged seasons between May and June.

The rainfall spatial trend analysis in the forward projection shows an increasing trend (>+0.0005 mm/day), except for a small area around the eastern Blue Nile basin, which increases by +0.0005 to +0.0010 mm/day. The majority part of NW Ethiopian highlands (western region) exhibits a fast-increasing trend (>+0.0025 mm/day), consistent with the time series analysis of the forward projection, which depicts an increasing trend of rainfall by +0.0023 mm/day (Figure 7c).

In Figure 8a, the spatial distribution of the past depicts a decreasing pattern from the south-central part (4.25 mm/day) of the region to the driest point in the northern extreme (1.75 to 2.25 mm/day), with modest rainfall along the western and eastern boundaries (2.25 to 3.25 mm/day) of the study area. However, most of the region has received sufficient rainfall between 3.25 and 4.75 mm/day, and the Shewa highlands (8–9° N and 37–38° E) around Wolkitie, with elevations between 2000 and 3500 m, are the wettest part (>5.25 mm/day). The mean annual rainfall spatial distribution (middle) of the past and change in the future (all units in mm/day)/right. Figure 8b shows the spatial distribution trend of wetter conditions (>+0.05 mm/day) in the study area, except for the Wollo highlands (east-central part) with no change. The greatest increase in wetter conditions (>+0.2 mm/day) is expected in the west-southern part of the region (including the Wollega and Benshagul highlands in the western Blue Nile basin). Additionally, in the south and west-south part of the highland regions, rainfall amounts between 4.5 and 5.6 mm/day are projected, indicating that this area will continue to have the highest rainfall potential, as confirmed by [54–57]. Thus, both the time series and spatial analysis of annual rainfall reveal an increase in rainfall in the NW Ethiopian highlands, consistent with the study by [58].

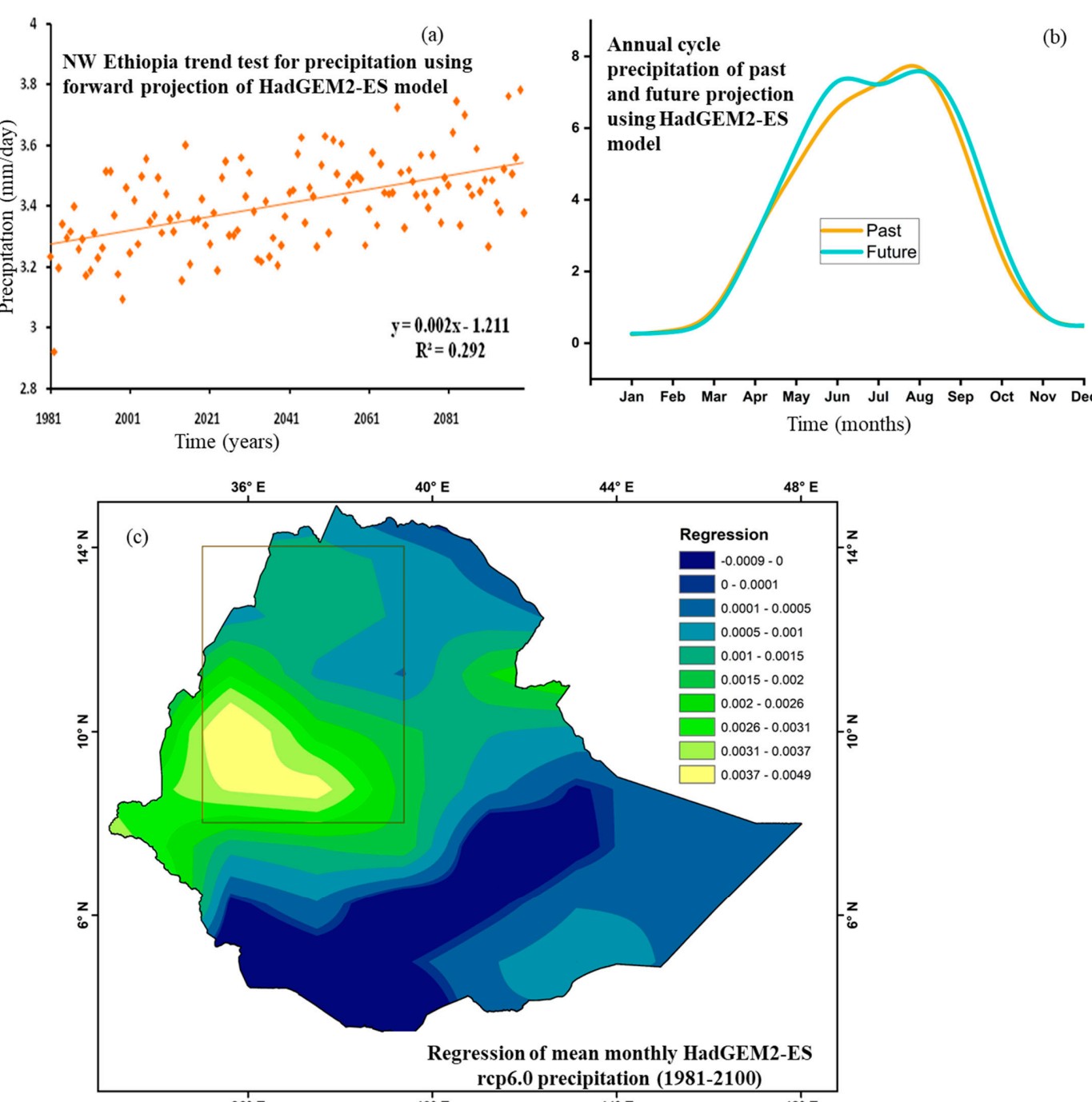

**Figure 7.** (**a**) Regression trend of rainfall forward projection from 1981 to 2100; (**b**) annual cycle precipitation monthly relative (%) curve from the HadGEM2-ES projection of past (1981–2010) and future (2041–2070); and (**c**) spatial trend of annual mean rainfall using Hadgem2-ES forward projection from 1981 to 2100.

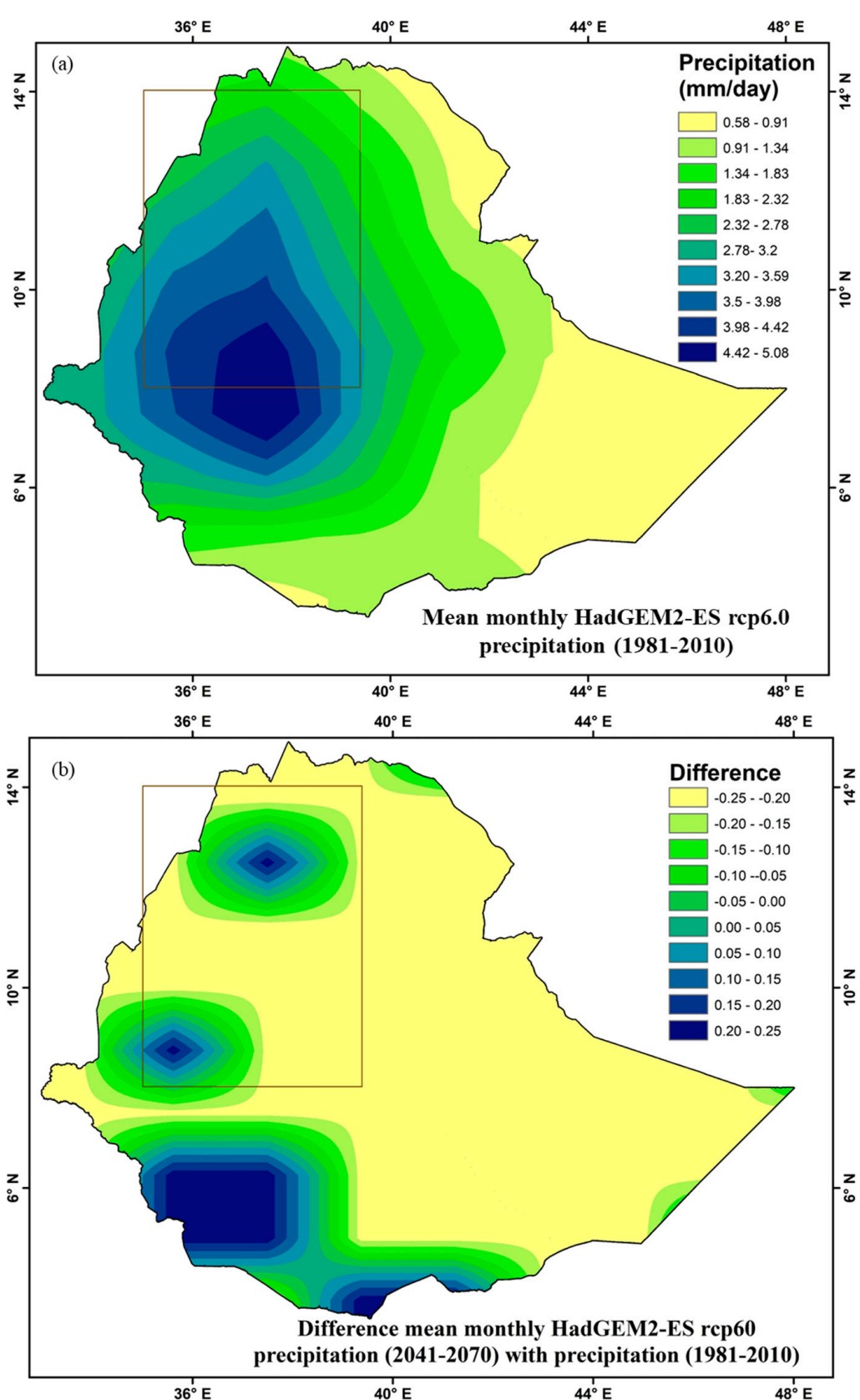

**Figure 8.** Spatial trend of annual mean rainfall using Hadgem2-ES for the past (**a**) and comparison of the difference between the past and the future (**b**).

### 3.3. Changes in Potential Evapotranspiration

PET is valuable for assessing moisture deficits and identifying periods with a high demand for irrigation and water harvesting. PET is derived from the surrogate sensible heat flux (SHF), which is obtained from the CMIP5 RCP6 HadGEM2-ES model's SHF output, adjusted with adjSHF (+2 mm/day). The annual PET cycle for both the past and the future indicates a mean of 3.64 mm/day, with a minimum of 3.62 mm/day and a maximum of 3.81 mm/day. Conversely, the mean for the future is 3.55 mm/day, with a minimum of 3.51 mm/day and a maximum of 3.70 mm/day. These values align with [37] suggestion of an average PET ranging from 3 mm/day to 5 mm/day for humid and sub-humid regions, with a mean daily temperature of around 20 °C.

The annual PET cycle pattern remains relatively stable from June to October, followed by an increase leading to the maximum during the hot season (February to April) (see Figure 9a). PET correlates with temperature, consistent with the findings of [59] in their study of Ethiopia's climate. To assess the change in PET from the past to the future, a trend test was conducted. The results indicate a decreasing trend of −0.0003 mm/day from 1981 to 2070 and an increasing trend of +0.0012 mm/day from 2041 to 2070. However, neither change is statistically significant, with *p*-values greater than 0.1517. Notably, PET exhibits a significant decreasing trend of −0.0036 mm/day in the past (1981–2010). Subsequently, all spatial maps in this chapter are presented in SHF (W/m2/month). To maintain consistency, the presentation is adjusted by converting SHF in W/m2 to PET/SHF in mm. The forward projection (1981–2070) regression trend spatial map for PET/SHF indicates minimal change (−0.0010 to +0.0010 mm/day) in most areas and a decreasing trend (−0.0010 to −0.0031 mm/day) in the north-western extreme (see Figure 9b). In general, the forward projection of the spatial trend aligns with the time series trend.

### 3.4. Changes in Soil Water Balance

The water balance technique, according to [60], involves quantitatively estimating the water circulation within the hydrosphere, lithosphere, and atmosphere. The soil water balance method is employed to calculate soil water balance by extracting precipitation (P) and PET (Es) data from the HadGEM2-ES CMIP5 rcp6 model for both the past (2041–2070) and the future. Before evaluating the water balance (P-Es) for the past and future, a trend analysis is conducted to determine the statistical significance of the change. The forward projection (2041–2070) and the past reveal a significantly increasing trend of +0.035 mm/day and +0.139 mm/day, respectively. However, the trend test for the future indicates a decreasing trend (−0.017 mm/day), but the change is not statistically significant. The computational procedures used to estimate changes in soil water content ($\Delta ST > 0$) are likely to yield reliable results regarding soil water content and its implications in the study area. The results show excess forcing ($\Delta ST > 0$) during the months of June to September (4 months) and deficit forcing ($\Delta ST \leq 0$) from October to May (8 months), both in the past and in the future (Table 4).

**Table 4.** Computation of monthly Acc. Pot. WL, soil water storage, soil water storage change, and actual evapotranspiration (mm/month).

| Time Period | Parameter | January | February | March | April | May | June | July | August | September | October | November | December |
|---|---|---|---|---|---|---|---|---|---|---|---|---|---|
| Past | Rainfall | 7 | 11 | 29 | 89 | 147 | 197 | 217 | 231 | 171 | 74 | 23 | 14 |
| | RO | 2.2 | 3.5 | 9.3 | 28.5 | 47.0 | 63.0 | 69.4 | 73.9 | 54.7 | 23.7 | 7.4 | 4.5 |
| (1981–2010) | PET/Es | 110 | 121 | 126 | 123 | 112 | 92 | 81 | 81 | 86 | 93 | 99 | 103 |
| | P-Es-RO | −105 | −114 | −106 | −62 | −12 | 42 | 67 | 76 | 30 | −43 | −83 | −93 |
| | ACC.P.WL | −325 | −439 | −545 | −607 | −619 | | | | | −43 | −126 | −220 |
| | ST | 38 | 22 | 13 | 9 | 8 | 50 | 117 | 193 | 200 | 161 | 105 | 66 |
| | ΔST | −28 | −16 | −9 | −4 | −1 | 42 | 67 | 76 | 7 | −39 | −56 | −39 |
| | AE (P + ΔST) | 35 | 27 | 38 | 93 | 112 | 92 | 81 | 81 | 86 | 93 | 79 | 53 |
| Future | Precipitation | 8 | 9 | 26 | 92 | 161 | 211 | 216 | 227 | 188 | 86 | 26 | 15 |
| | RO | 2.6 | 2.9 | 8.3 | 29.4 | 51.5 | 67.5 | 69.1 | 72.6 | 60.2 | 27.5 | 8.3 | 4.8 |
| (2041–2070) | PET/Es | 123 | 140 | 152 | 139 | 111 | 94 | 84 | 82 | 84 | 88 | 97 | 108 |

**Table 4.** *Cont.*

| Time Period | Parameter | January | February | March | April | May | June | July | August | September | October | November | December |
|---|---|---|---|---|---|---|---|---|---|---|---|---|---|
| | P-Es-RO | −118 | −134 | −134 | −76 | −2 | 49 | 63 | 72 | 44 | −30 | −79 | −98 |
| | ACC.P.WL | −325 | −459 | −593 | −669 | −671 | | | | | −30 | −109 | −207 |
| | ST | 39 | 20 | 10 | 7 | 7 | 56 | 119 | 192 | 200 | 172 | 115 | 70 |
| | ΔST | −31 | −19 | −10 | −3 | 0 | 49 | 63 | 72 | 8 | −28 | −57 | −45 |
| | AE (P + ΔST) | 39 | 28 | 36 | 95 | 111 | 94 | 84 | 82 | 84 | 88 | 83 | 60 |

Ro = runoff, PET = potential evapotranspiration, Es = actual evapotranspiration, ACC.P.WL = accumulated potential water loss, ST = soil moisture storage, ΔST = soil moisture change.

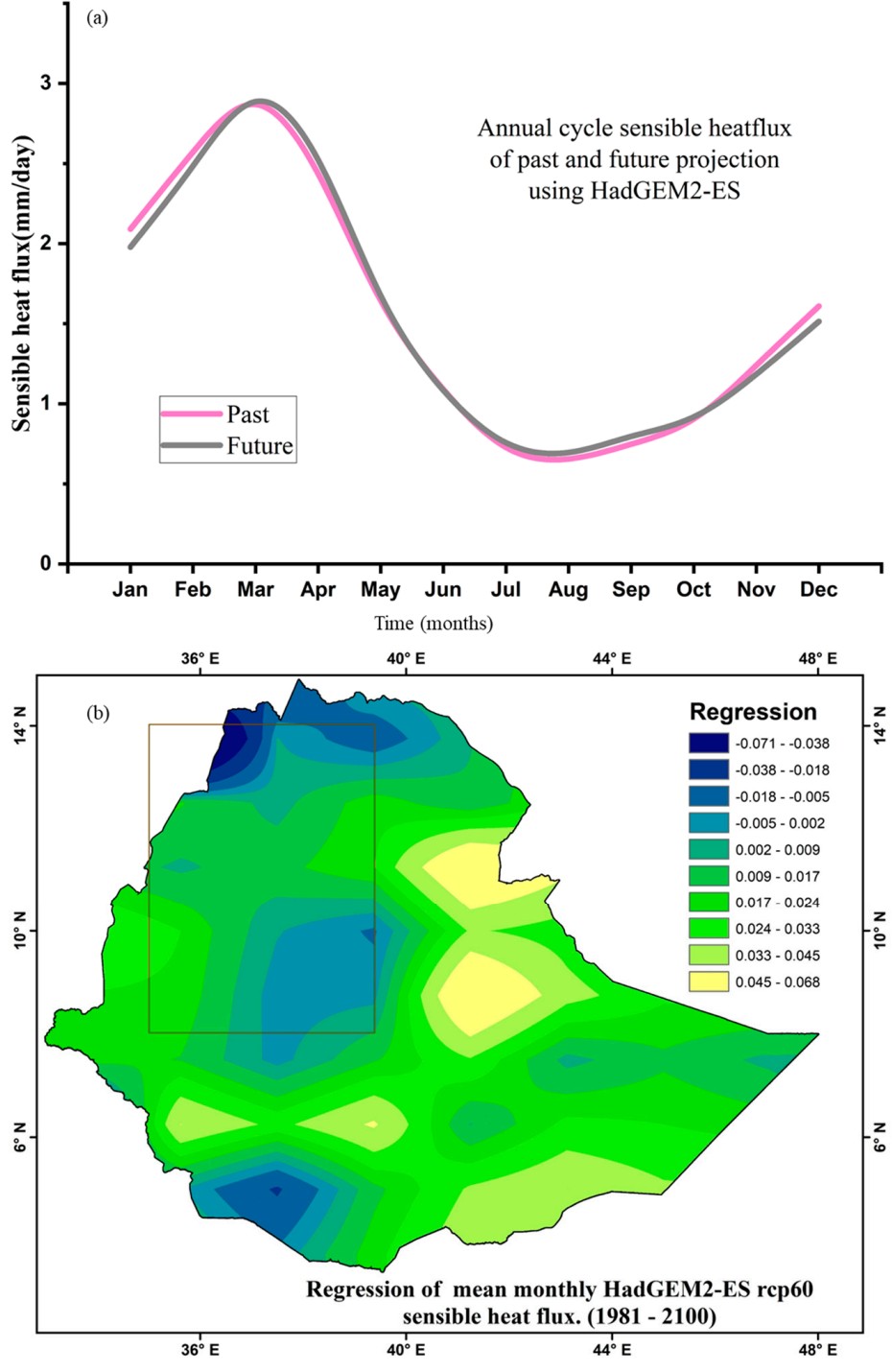

**Figure 9.** Annual cycle of sensible heat flux of the past (1981–2010) and the future (2041–2070) (**a**), and spatial trend of sensible heat flux using Hadgem2-ES for the past (**b**).

*3.5. Variability in Length of Growing Season*

The annual cycle of soil water change (ΔST) was computed for both past and future years. In the NW Ethiopian highlands, estimates of the average reserve of soil water storage were 82 mm/month in the past and a projected 84 mm/month in the future. Both are positively affected by ΔST > 0, starting from the onset day of the year (DOY). However, soil water content changes from the first day (offset DOY) (ΔST < 0) influence the driving soil water content to become deficient. The trend analysis test revealed a decreasing trend in onset date (−0.021 day/yr) and cessation date (−0.300 day/yr) in the past. Still, LGP showed an increasing trend (+0.033 day/yr). In the future, there is a decreasing trend in cessation date (−0.039 day/yr) and LGP (−0.335 day/yr), while the onset date shows an increasing trend (+0.295 day/yr). However, these changes are not statistically significant for any of the parameters at a 0.05 significance level. The statistical median result for the onset date was 139 DOY, for the cessation date it was 263 DOY, and for LGP it was 130 days in the past. In the future, the onset date, cessation date, and LGP are likely to be 137 DOY, 270 DOY, and 136 days, respectively.

Our results align with [61], who emphasized the significant variability in seasonal rainfall, especially during the cropping season, potentially resulting in moisture stress for maize plants and impacting the LGP. A slightly similar result for LGP, ranging between 137 and 205 days for the NW Ethiopian highlands, has been reported in the mapping of climate vulnerability and poverty in Africa (current condition of 2000) by [62,63] also suggested a planting date in April and an LGP of 180 days for maize (grain) in East Africa's high-altitude areas. The LGP estimation obtained in both time periods has a minimum LGP ≥ 118 days, which is above the threshold LGP (90 days). Ref. [39], most crops grown in Ethiopia, with the exception of some pulses and very low-yielding varieties of Teff (a common cereal food in Ethiopia) and wheat, require a growing period of at least 90 days. Additionally, ref. [64] recommended a 90-day LGP as the optimum LGP in the case of Ethiopia. Figure 9 illustrates the evaluation of the variability of the length of growing seasons, showing highly variable onset dates, followed by less variable LGP, and offset dates in the past. In the future, offset dates are likely to be highly variable, while LGP and onset dates are less variable. In general, the evaluation of the onset date, end date, and LGP of crop growing seasons under a climate change scenario emphasizes the importance of early planting in May and harvesting at the end of September, with LGP suitable for single and, to some extent, double cropping both in the past and future. There is a slightly longer growing season in the future, which is likely to benefit agricultural production in the NW Ethiopian highlands.

## 4. Conclusions and Summary

This study provides in-depth analyses of climate change and its impact on soil water balance and the length of growing seasons in the NW Ethiopian highlands. Utilizing data sets from station observations, gridded observations, reanalysis, and simulations of CMIP5 rcp6 models, various analysis tools, including trends, spatial maps, and measures of bias and dispersion, were employed for a comprehensive result analysis. The validation criteria identified CCSM4 (for tasmax) and HadGEM2-ES (for precipitation and PET/adjshf) as the models meeting most criteria, serving as data sources for climate change, variability, extremes, and soil water balance analysis.

Temperature exhibits a significant increasing trend across all periods, with tasmax increasing faster than tasmin, and the hot season surpassing the cold season in both historical and future contexts. The spatial maps for tasmax and tasmin projections indicate a faster warming trend in the region. The rainfall trend analysis in the NW Ethiopian highlands is mixed, with significant increases in JJAS season across all periods and non-significant trends in other seasons and annual rainfall. The spatial distribution aligns with the time series trend, suggesting a wetter future, particularly in the southwest.

The extreme decrease in minimum rainfall necessitates substantial adjustments in agriculture and water management. Conversely, the maximum rainfall changes benefits agriculture, except in flood-prone areas. The extreme probabilities for the FMAM season suggest that it may not be viable for rain-fed crops alone but could support early planting. The ONDJ season, anticipated to bring substantial rainfall, could serve as an additional growing season, or it could extend JJAS, but early warning is essential to avoid disruptions during harvest.

PET trends are mixed, with significant changes only in the past. Comparisons with rainfall and temperature indicate low PET rates during low temperatures and vice versa. The water balance estimation reveals a significant increasing trend in the past but a non-significant decreasing trend in the future. The annual total water balance conditions show PET exceeding rainfall in both periods.

This study underscores the importance of early planting in April/May and harvesting in September, suitable for single and, to some extent, double cropping in the past and future. The growing season variability indicates highly variable LGP dates, with onset and offset dates less variable in the past. In the future, onset dates are expected to be highly variable, while LGPs and offset dates remain less variable. Overall, this study suggests crucial implications for farming in the NW Ethiopian highlands. Farmers should adapt by adjusting planting and harvesting times, benefiting from the projected increase in rainfall. Efficient water management, particularly in irrigation, can capitalize on the observed rise in soil water balance, enhancing agricultural resilience. Diversifying crops and introducing new varieties aligns with the anticipated lengthening of the growing season, optimizing productivity. Integrating these findings into climate-resilient planning is essential for regional authorities, with a focus on sustainable water resource management and infrastructure development. Ongoing research remains vital for refining adaptation strategies and staying abreast of evolving climate dynamics.

**Supplementary Materials:** The following supporting information can be downloaded at: https://www.mdpi.com/article/10.3390/cli11120243/s1, Figure S1: Flow chart of how the research steps interlink is illustrated. Table S1: Runoff coefficient: empirical coefficient representing fraction of rainfall that runs off (adapted from [42]). Table S2: Summary for the methods how the research is performed.

**Author Contributions:** G.B.T.: Conceptualization, Methods, Supervision, Analysis, Writing—Original Draft, Writing—reviewing & editing. A.A.A.: Conceptualization, Methods, Data extraction, Data analysis (Figures, Tables), Writing—Original Draft, Writing—reviewing & editing. S.E.D.: Overall supervision, conceptualization, Writing—review & editing. All authors have read and agreed to the published version of the manuscript.

**Funding:** This research received no external funding.

**Data Availability Statement:** The data presented in this article are provided based on the inquiries related to the data may be directed to Gashaw Bimrew Tarekegn, and Addis A. Alaminie at gashbimrew@gmail.com or metaddi@gmail.com.

**Conflicts of Interest:** The authors declare no conflict of interest.

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
