# Peer review of "Linking Climate Change Information with Crop Growing Seasons in the Northwest Ethiopian Highlands"

_climate, doi:10.3390/cli11120243_

Round 1

Reviewer 1 Report

Comments and Suggestions for Authors

The manuscript evaluates the impact of climate change on cropping season in Ethiopia's highlands. The authors used CMIP5 climate projections along with validated reanalysis historical weather data. My comments about the manuscripts are stated below:

Major Revisions

1) We will advise the author to present the seasonal analysis with the standard meteorological season (DJF, MAM,JJA, SON). This will allow global readers to connect with the article. Special seasons relevant for the region may also be presented.  

2) The reanalysis data was calibrated and validated with an area average of 35 stations over the mountainous region. This method is not appropriate because high variability in rainfall and temperature in the region will create a lot of errors. We propose a comparison of reanalysis datasets with observed at each meteorological station. An interpolated average of the result will give more clarity to the subject.

3) Authors have used r and R2 for comparing simulated and observed data. We propose the use of more than two standard indices. the NSE, KGE and RMSE are recommended for this type of study. 

4) The graphics presented are arranged in a confusing manner. Author should properly name the graphics and arrange them properly. 

 5) Authors should provide the method for calculation of runoff (RO) in the study.

6) Authors should show the location of all the 35 meteorological stations on the highlands. 

Author Response

Response to reviewers’ comments (climate-2693224)

 We thank the editor and the reviewers for their constructive and valuable comments, and for providing us an opportunity to revise our manuscript. We have carefully considered all the comments and made the necessary changes to our revised manuscript. Our detailed responses to these comments are provided below.

We have numbered all the comments for ease of cross-referencing. The in the revised manuscript shows the changes done in track change mode to our original submission. The italic text in responses below shows the new text added/edited in response to the reviewers’ comments in the revised manuscript.

 Editor’s comments

Based on the reviewers' reports and decisions I also recommend a major revision of your manuscript. I suggest that you respond and pay attention to all reviewers' suggestions, recommendations, and questions in order to produce a new version, with the necessary quality so that it can be accepted for publication.

Response:

We thank the editor for providing us an opportunity to respond to the reviewers’ comments and revise our manuscript accordingly. As suggested, we have carefully addressed all the reviewers' comments that have helped us to substantially improve the quality of our manuscript. In response to comments below, the revisions have also been made throughout the manuscript, starting from abstract through to conclusions. Our detailed responses to reviewers’ comments and the associated changes made to the manuscript are given below. 

Reviewer #1

The manuscript evaluates the impact of climate change on cropping season in Ethiopia's highlands. The authors used CMIP5 climate projections along with validated reanalysis historical weather data. My comments about the manuscripts are stated below:

Response:

We thank the reviewer for time and useful inputs to improve the manuscript.

1.1 We will advise the author to present the seasonal analysis with the standard meteorological season (DJF, MAM, JJA, SON). This will allow global readers to connect with the article. Special seasons relevant for the region may also be presented. 

Response:

Thank you and the reasons are justified as follows: “The National Meteorology Agency (NMA) has classified and recommended applying the seasons in Ethiopia into three periods: the rainy season from June to September, the dry season from October to January, and the short rainfall season from February to May, as documented by Workineh (1987), Haile (1986), and NMA (1996). In alignment with this classification, our manuscript categorizes the seasons into both two and three. Specifically, the seasons in our study are primarily two: the rainy season from May to October and the dry season from November to April.” 

1.2  The reanalysis data was calibrated and validated with an area average of 35 stations over the mountainous region. This method is not appropriate because high variability in rainfall and temperature in the region will create a lot of errors. We propose a comparison of reanalysis datasets with observed at each meteorological station. An interpolated average of the result will give more clarity to the subject.

Response:

We appreciate the reviewer's concern regarding the calibration and validation of the reanalysis data. It's important to note that the reanalysis data and results were conducted using a grid box with a spatial resolution of 50km x 50km. In this context, each grid box typically encompasses three to six meteorological stations based on their respective locations, rather than a single point station per grid box. Given that the rainfall system is predominantly influenced by global patterns, the impact of local factors such as topography and land use on rainfall fluctuations is minimal. This results in lower uncertainty and error in the presented findings. While we acknowledge the suggestion for a comparison with observed data at each meteorological station, the grid-based approach provides a comprehensive perspective on the regional dynamics and trends, contributing to a robust understanding of the subject matter.

1.3 Authors have used r and R2 for comparing simulated and observed data. We propose the use of    more than two standard indices. the NSE, KGE and RMSE are recommended for this type of study. 

Response:

Thank you for your valuable comment. In our study, we employed a stepwise evaluation method, as recommended by Mark R. J. (2014) and Schwalm et al. (2013). This approach systematically evaluates and compares the observed data with simulated data, taking into account not only overall fit but also the consideration of seasonal patterns and peak occurrences. While we acknowledge the suggestion to include more than two standard indices such as NSE, KGE, and RMSE, we believe that our chosen method aligns well with the objectives of our study and allows for a comprehensive evaluation of climate change models. This methodological choice was made with the aim of selecting the most appropriate model to capture the complexities of the climate system in our analysis.

1.4  The graphics presented are arranged in a confusing manner. Author should properly name the graphics and arrange them properly. 

Response:

Thank you for this comment. The section is now revised, and we believe that we named the graphics properly and arrange them properly in the entire article.

1.5   Authors should provide the method for calculation of runoff (RO) in the study.

Response:

Thank you for your positive feedback for our workk and revised as follows and included in the manuscript that read as: “RO (Runoff) will be calculated as follows: Equation:  . Where Cor (runoff coefficient) by estimating  Cro for each site based on the soil type steep and range of the slope (McBean et al., 1995).

Table S1. Runoff coefficient: empirical coefficient representing fraction of rainfall that runs off (adapted from McBean et al., 1995).

Runoff coefficients for landfill

Surface Conditions: Grass cover (slope)

Runoff Coefficient

Sandy soil, flat, 2%

0.05 - 0.10

Sandy soil, steep, 7%

0.10 - 0.15

Heavy soil, flat, 2%

0.15 - 0.20

Heavy soil, average, 2-7%

0.13 - 0.17

Heavy soil, steep, 7%

0.18 - 0.22

Sandy soil, average, 2-7%

0.25 - 0.35

1.6 Authors should show the location of all the 35 meteorological stations on the highlands. 

Response:

Thank you for the comment we have already incorporated the location of the stations as the map in Figure 1.

Figure 1. Shows the location of the 35 meteorological stations in the highlands of Ethiopia.

Reviewer 2 Report

Comments and Suggestions for Authors

In the article “Linking Climate Change information with Crop Growing Seasons in the Northwest Ethiopian Highlands” the future climate conditions of NW Ethiopian highlands are investigated. The authors used CMIP5 model data. The study is important because climate change is affecting crop production in many regions of the world, including Ethiopia.

The article requires major revision.

Comments

The article is poorly formatted, figures and tables are not inserted into their proper places, there are many errors – all this makes reading difficult.

The authors choose RCP6.0. The work does not say why the authors chose this particular scenario.

Lines 140-141: the sentence is incomplete, please rewrite

Lines 153-154: this link is not in the Reference list and there is no year

Authors should decide on data sources

Lines 180-182: it is stated that the MERRA data has poor correlation with PET station data, so you use ERA-Interim, but then (line 236) you mention MERRA again for PET calculation. For what?

Lines 182-183: after mentioning the European Centre for Medium-Range Weather Forecasts you need to write ERA-Interim, because on line 187 you mention ERA-Interim

Lines 252, 262, 263: where are these sections and equation?

Table S1: on lines 310-311 it is written that in the table the summary for the methods, but there are climatic parameters

Line 324: EPCC (2015)? Maybe it should be IPCC?

Line 381: not figure 6

Lines 391-392 Figure 7 (middle): it is written “the main changes in May-June and Sept-Oct”. What is the difference between the curves for these months? it is obvious that it is statistically insignificant

Lines 402-410: Figure 6 left. I don’t see such trend values and their location in the figure

Line 433: what figure number should be here?

Line 430: what “table 3.1 below” mean?

Lines 531-533: On what basis is this conclusion made?

Most references are later than 2013-2014. Modern references should be added

Figure 1. There is no link to this work in the references list, the full year is not indicated

Figure 2: why is this figure shown? what do the lines on it mean?

Figure 3: no axes labels, abbreviations must match the text of the article (tasmax and tasmin)

Figures 4-8:  The significance of trends is not shown on spatial maps

The quality of all figures should be improved.

The main comment: Apart from air temperature, the remaining parameters did not show statistically significant differences between the past and the future. Therefore, it is doubtful to draw any conclusions about the improvement or deterioration of conditions for crop production in Ethiopia.

Author Response

Reviewer #2

In the article “Linking Climate Change information with Crop Growing Seasons in the Northwest Ethiopian Highlands” the future climate conditions of NW Ethiopian highlands are investigated. The authors used CMIP5 model data. The study is important because climate change is affecting crop production in many regions of the world, including Ethiopia.

Response:

Thank you for your positive feedback and summary of our work.

2.1    The article is poorly formatted, figures and tables are not inserted into their proper places, there are many errors – all this makes reading difficult.

Response:

Thank you for bringing this to our attention. We have carefully revised the article, addressing the formatting issues. Figures and tables have been appropriately inserted into their designated places, and we have ensured that they are named accurately. These revisions aim to enhance the overall readability and coherence of the article.

2.2  The authors choose RCP6.0. The work does not say why the authors chose this particular scenario.

Response:

Thank you for your insightful comment regarding our choice of the RCP6.0 scenario. We have carefully revised the manuscript to address this concern, and the following information has been incorporated in Section 2.2. The Royal Netherlands Meteorological Institute (KNMI) serves as the host for the Climate Explorer website, providing access to CMIP5 data. KNMI collaborates closely with international meteorological networks, including the European Centre for Medium-Range Weather Forecasts (ECMWF) and the European meteorological satellite network EUMETSAT. The organization of KNMI Climate Explorer (CE) by Touret and Oldenborgh (2013) is highlighted, and Table S2 provides a list of CMIP5 model groups available to the public on KNMI CE. In selecting the specific scenario, this study focused exclusively on CMIP5 RCP6 models’ data sources developed by the AIM modeling team at the National Institute for Environmental Studies (NIES) in Japan. The rationale for this choice is outlined as follows:

  1. Stabilization Scenario – CMIP5 RCP6 represents a stabilization scenario wherein total radiative forcing is stabilized shortly after 2100, without overshooting. This is achieved through the application of various technologies and strategies for reducing greenhouse gas emissions (Fujino et al., 2006, and Hijioka et al., 2008). Given the global consensus on the imperative to reduce carbon emissions and mitigate climate change, RCP6 was deemed a relevant scenario for our study.
  2. Model Diversity – CMIP5 RCP6 models comprise 23 to 25 models, serving as data sources for 21 climate variables. All model outputs were considered with equal weight, acknowledging the diversity resulting from different model versions and parameterizations. Further details on each CMIP5 RCP6 model are referenced to Taylor et al. (2012) and the website cmip-pcmdi.llnl.gov/cmip5.
  3. Validation in Ethiopian Highlands – Previous studies conducted in the Ethiopian highlands by Becker et al. (2013) and Jury (2015) confirmed minimal bias when using CMIP5 RCP6 models. Additionally, the average CMIP5 RCP6 models’ data exhibited a significant annual cycle correlation with GPCC V6 rainfall (r ≥ 0.6) and CRU TS v3.22 maximum temperature (r ≥ 0.6) in the Ethiopian highlands, demonstrating the reliability of RCP6 in capturing rainfall and temperature dynamics.

Considering the above justifications and the limitations of our research capacity, we chose to use CMIP5 RCP6 model data sources to project the climatic conditions in the NW-Ethiopian highlands. It is essential to note that the results presented in this study are specific to the CMIP5 RCP6 scenario. Nonetheless, we acknowledge the importance of exploring the remaining three scenarios when assessing broader climate impacts.

2.3 Lines 140-141: the sentence is incomplete, please rewrite.

Response:

Thank you for your comment: It is rewritten as on the other hand; the Ethiopian highlands soil type is dominantly Vertisols account for 12.7 million hectares in Ethiopia of which 7.6 minion hectares is in the NW Ethiopian highlands.

2.4 Lines 153-154: this link is not in the Reference list and there is no year

Response:

Thank you for your comment: The year for citation is included and listed in the reference list.

2.5 Authors should decide on data sources. Lines 180-182: it is stated that the MERRA data has poor correlation with PET station data, so you use ERA-Interim, but then (line 236) you mention MERRA again for PET calculation. For what?

Response:

Thank you for your comment:  it is mentioned just to remind that MERRA data source was used for the comparison purpose.

2.6 Lines 182-183: after mentioning the European Centre for Medium-Range Weather Forecasts you need to write ERA-Interim, because on line 187 you mention ERA-Interim

Response:

Thank you for your comment it is rewritten as Thus, European Centre for Medium-Range 183---Weather Forecasts (ECMWF) ERA-Interim adjshf served as the validation reference for CMIP5 rcp6 sensible heat flux, after adjustment to match ECMWF adjshf.

2.7 Lines 252, 262, 263: where are these sections and equation?

Response:

Thank you “cf: section 2.17.1 is deleted.

2.8  Table S1: on lines 310-311 it is written that in the table the summary for the methods, but there are climatic parameters.

Response:

Thank you for the comment and the climate parameters are deleted and replaced by the summary of the methods.

 2.9 Line 324: EPCC (2015)? Maybe it should be IPCC?

Response:

Thank you for the comment: EPCC is an abbreviation to “Ethiopian Panel for Climate Change.”

2.10 Line 381: not figure 6.

Response:

Thank you for the comment: It is corrected to figure 7a.

2.11     Lines 391-392 Figure 7 (middle): it is written “the main changes in May-June and Sept-Oct”. What is the difference between the curves for these months? it is obvious that it is statistically insignificant?

Response:

Thank you for the comment: The future projection of rainfall shows a slight increase at the months of May-June and Sept-Oct. We thought 0.5mm/day is 15mm/month and 30mm for the two months that may have some potential of destruction special during the harvesting time.

2.12      Lines 402-410: Figure 6 left. I don’t see such trend values and their location in the figure

Response:

Thank you for the comment: We think you mean Figure 8 (The figures on figure 6 are about temperature)? The value to south-central Ethiopia is corrected to 4.25mm/day.

2.13      Line 433: what figure number should be here?

Response:

Thank you for the comment: It was wrong numbering and corrected as figure 9 left top.

2.14     Line 430: what “table 3.1 below” mean?

Response:

Thank you for the comment: It was wrong numbering and corrected in Table 4.  

2.15      Lines 531-533: On what basis is this conclusion made?

Response:

Thank you very much and now it is deleted.

2.16     Most references are later than 2013-2014. Modern references should be added

Response:

Thank you for your valuable feedback. As suggested, we have incorporated recent citations (2020 – 2023) to enhance the relevance of our manuscript. Some of the notable additions include:

  1. Cramer, W., Guiot, J., Marini, K., Azzopardi, B., Balzan, M.V., Cherif, S., Doblas-Miranda, E., Dos Santos, M.J.P.L., Drobinski, P., Fader, M., and Hassoun, A.E.R. (2020). "MedECC 2020 summary for policymakers. Climate and Environmental Change in the Mediterranean Basin–Current Situation and Risks for the Future. First Mediterranean Assessment Report," pp. 11-40.
  2. Yonggang, G.A.O., Ming, W.A.N.G., JIANG, L., Fang, Z.H.A.O., Feng, G.A.O., and Huiying, Z.H.A.O. (2023). "Dynamics of carbon budget and meteorological factors of a typical maize ecosystem in Songnen Plain, China." Spanish Journal of Agricultural Research, 21(4), pp. e0301-e0301.
  3. Hultine, K.R., Hernández-Hernández, T., Williams, D.G., Albeke, S.E., Tran, N., Puente, R., and Larios, E. (2023). "Global change impacts on cacti (Cactaceae): current threats, challenges and conservation solutions." Annals of Botany, p. mcad040.
  4. Horgan, F.G. (2020). "Potential for an impact of global climate change on insect herbivory in cereal crops." Crop protection under changing climate, pp. 101-144.
  5. Lesk, C., Anderson, W., Rigden, A., Coast, O., Jägermeyr, J., McDermid, S., Davis, K.F., and Konar, M. (2022). "Compound heat and moisture extreme impacts on global crop yields under climate change." Nature Reviews Earth & Environment, 3(12), pp. 872-889.
  6. Senbeta, A.F., and Worku, W. (2023). "Ethiopia’s wheat production pathways to self-sufficiency through land area expansion, irrigation advance, and yield gap closure." Heliyon.
  7. Markos, D., Worku, W., and Mamo, G. (2023). "Spatio-temporal variability and rainfall trend affects seasonal calendar of maize production in southern central Rift Valley of Ethiopia." PLOS Climate, 2(6), p. e0000218.
  8. Van Vuuren, D.P., Edmonds, J., Kainuma, M., Riahi, K., Thomson, A., Hibbard, K., Hurtt, G.C., Kram, T., Krey, V., Lamarque, J.F. and Masui, T., 2011. The representative con-centration pathways: an overview. Climatic change, 109(1-2), p.5.

 2.17      Figure 1. There is no link to this work in the references list, the full year is not indicated.

Response:

Thank you for your comment. We have corrected the year, and Figure 1 has been replaced with a new Figure illustrating the location of the study area and the distribution of the meteorological stations used.

2.18     Figure 2: why is this figure shown? what do the lines on it mean?

Response:

Thank you for your comment. Our objective was to provide an explanation of CO2 under the CMIP5 RCP6 scenario. However, as we clarified the reasons for selecting the CMIP5 RCP6 scenario in Section 2.2, this explanation was largely redundant, and we have subsequently removed it.

2.19     Figure 3: no axes labels, abbreviations must match the text of the article (tasmax and tasmin).

Response:

Thank you very much and we have revised and tried to much the abbreviations of the article with the figures.  

 2.20    Figures 4-8:  The significance of trends is not shown on spatial maps.

Response:

Thank you for the comment: Figure 4 (Tmin and Tmax) and Figure 7 (Rainfall) have spatial map to explain about the trends that is presented in section 3.1 and 3.2 in the manuscript.

2.21     The quality of all figures should be improved.

Response:

As suggested now, now we have improved all the Figures and amended in lines xx-yy.

2.22       The main comment: Apart from air temperature, the remaining parameters did not show statistically significant differences between the past and the future. Therefore, it is doubtful to draw any conclusions about the improvement or deterioration of conditions for crop production in Ethiopia.

Response:

Thank you for your positive summary of our work. In addition to temperature, annual rainfall, and the seasons from June to September (wet season) and October to January (dry season) were statistically significant. These two seasons play a crucial role in crop production in the NW-Ethiopian highlands.

Reviewer 3 Report

Comments and Suggestions for Authors

If I understand correctly, this study looks at the impacts of climate change on the LGS in the NE Ethiopian Highlands. Climate variables are Tmax, Tmin, precip, PET, and soil moisture. Two models are taken from the CMIP5 dataset based on how well they agree with observations from the National Met. Agency.  Although this is a potentially very interesting study, I do not think it is presently fit for publication. I recommend the paper be rejected I its present form but that the authors be encouraged to resubmit once the following points are addressed.

The overriding trouble with the paper is the lack of clarity. This is mostly due to the grammar and syntax but also conflation of different concepts, poor figure captions and annotation, and also inconsistent naming. In some parts of the paper, I still have difficulty understanding what exactly the authors are writing about. I accept that I am not an agronomist and a lot of my lack of understanding may be due to my personal limitations, but the paper should be accessible to climate impact scholars like myself. Many of the figures also need to be fixed for unclear or missing axis titles, improper scaling, etc.

I have taken the liberty of rewriting the first paragraph of the abstract were I think “risks” is used for “impacts”, “anticipated to” is used for “expected to”, “climate change trends” should be “climate trends”, etc..

-line 11 to 19)

 Abstract (possible rewrite)

“In Ethiopia, climate change impacts are expected to have significant consequences for agriculture and food security. This study investigates past (1981-2010) and future (2041-2070) climate trends and their influence on the length of growing season (LGS) in the North-Western Ethiopian Highlands. Climate observations were obtained from the National Meteorological Agency of Ethiopia while the best performing and highest resolution models from the CMIP5 experiment and RCP6 (Coupled models Intercomparison Project and representative concentration pathway 6) were used for the analysis. Standard statistical methods are applied to compute soil water content as well as to evaluate climate variability and trends along with their impact on LGS.”

Below is a list of points in the paper that need to be addressed. The list is by no means exhaustive (I stopped noting every issue I found after line 150) and I encourage the authors to have the paper professionally proofread.  

-Line 19 to 21) I do not understand what point the authors are trying to make here

-line 19 to 23) suggested rewrite: “In the area of study maximum daily temperatures rose, from 1980-2010 to 2040-2070, by 1.3C while minimum daily temperatures rose by only 1.2C.

-line 22) -1.2C should be +1.2 C

-line 24 to 26) whereas, and, of ….  Does not make sense...

-line 28) “to past condition,” this should be a period.

-line 29) the terms “projected” is used in an unclear way throughout the paper. I think, though I am not certain, that “projected” refers to the type of simulation run taken from the CMIP5 data set. Hence there is a “projected past” and a “projected future”.  If so, this is not made clear in the text. For example, in line 318, the authors talk about “all time periods (past, future and forward projection) “. What time period is the “forward projection”.

-line 45) what does “This” refers to?  This… what?

-line 60 and line 74) Both paragraphs claim to address the effects of growing temperature. The paragraph at line 60 starts “The increase in temperature due to climate change has both positive and negative impacts..”, and the one at line 74 starts “An increase in average temperature can have several effects…”

These two paragraphs should be merged.

-lines 93-95) This sentence has no verb.

-line 98) “.. total crop failure resulted the high-…”. May I suggest: “..to total crop failure. This caused the high-…”

107 to 110) “Moreover, the impact analysis of climate change on soil water content and the onset and cessation date and length of growing period derived from soil water content mostly provides reliable result than other methods used to determine crop growing seasons.” May I suggest: “Moreover, the impact analysis of climate change on soil water content, and the onset and cessation date and length of growing period derived from soil water content, provides a more reliable result than other methods used to determine crop growing seasons.”

-line 114) “the past (1981-2010) and future (2041-2070) CMIP5 rcp6 model projections comparative observations..”. I do not understand this nomenclature. What is a “past model projection comparative observation..”?

-line 126) as “a” source

-line 127) “peasant agriculture has polyculture” change to “peasant agriculture is a polyculture”

-line 129) “meets” not “met”

-line 132) “contributed for the” change to “contributed to the…”

-line 140) “dominantly Vertisols account” to “dominantly Vertisols and accounts”

-line 141) “of which 7.6 minion hectares.”  What does this section of sentence mean?

Other issues:

-line 210) CO2 should be CO2

-line 264) “steep with soil ranging”?

-line 351 to 356) these detailed descriptions of the temp change in each plot are not helpful to the paper

Figure 2) what is the blue line? CO2 should be CO2.

Figure 3) Unclear Y-axis title. Also Y-axis ranges, or at least the scale, in panel “b”, and ”c” should be the same. Also, the fonts of the annotations are different.  

Figure 4) In all the figures of this kind, showing a plot of the country, the scales are different. Not only are the figures different sizes, but the latitude and longitude scales change (the country changes shape).

Figure 4)   In all the figures of this kind, the titles are the original titles given by the software used. These titles should be changed so the reader is not expected to understand all of the acronyms of the software.

Figure 4) In all the figures of this kind, the units for the contour levels are often missing.

Figure 4) I do not understand how this figure is a “spatial trend projection”. Perhaps it is a “spatial distribution of temperature trends”. I am not sure because I do not see any units, but the numbers look like Degrees/year.

Figure 5) panel “b” looks like a “spatial distribution of the temperature change” not a “spatial

change in the future.”

Figure 6 to 8) See all points above.

Figure 9) How is this a trend? In the text says “variability”. Also, I do not understand why the caption refers to rainfall and the plot y-axis (panels a and b) refer to PET and hfss.  Also, these figures are pasted over other figures that are showing underneath.

Comments on the Quality of English Language

See review.

Author Response

Response to Reviewer #3

If I understand correctly, this study looks at the impacts of climate change on the LGS in the NE Ethiopian Highlands. Climate variables are Tmax, Tmin, precip, PET, and soil moisture. Two models are taken from the CMIP5 dataset based on how well they agree with observations from the National Met. Agency.  Although this is a potentially very interesting study, I do not think it is presently fit for publication. I recommend the paper be rejected I its present form but that the authors be encouraged to resubmit once the following points are addressed.

Response:

Thank you for your positive summary of our work. As suggested, we have revised the manuscript.

3.1    The overriding trouble with the paper is the lack of clarity. This is mostly due to the grammar and syntax but also conflation of different concepts, poor figure captions and annotation, and also inconsistent naming. In some parts of the paper, I still have difficulty understanding what exactly the authors are writing about. I accept that I am not an agronomist and a lot of my lack of understanding may be due to my personal limitations, but the paper should be accessible to climate impact scholars like myself. Many of the figures also need to be fixed for unclear or missing axis titles, improper scaling, etc.

Response:

Thank you for your positive feedback.

3.2    I have taken the liberty of rewriting the first paragraph of the abstract where I think “risks” is used for “impacts”, “anticipated to” is used for “expected to”, “climate change trends” should be “climate trends”, etc,  (line 11 to 19).

Response:

We appreciate the positive inputs to our manuscript.  

3.3    Abstract (possible rewrite).“In Ethiopia, climate change impacts are expected to have significant consequences for agriculture and food security. This study investigates past (1981-2010) and future (2041-2070) climate trends and their influence on the length of growing season (LGS) in the North-Western Ethiopian Highlands. Climate observations were obtained from the National Meteorological Agency of Ethiopia while the best performing and highest resolution models from the CMIP5 experiment and RCP6 (Coupled models Intercomparison Project and representative concentration pathway 6) were used for the analysis. Standard statistical methods are applied to compute soil water content as well as to evaluate climate variability and trends along with their impact on LGS.”

Response:

Thank you very much for the comment and we have revised inline to the comment in the manuscript.

3.4    Below is a list of points in the paper that need to be addressed. The list is by no means exhaustive (I stopped noting every issue I found after line 150) and I encourage the authors to have the paper professionally proofread.

 Response:

We thank the reviewer for this valuable suggestion. As suggested by the reviewer we have employed professional proofreader and revised the whole manuscript.

 3.5      Line 19 to 21) I do not understand what point the authors are trying to make here

 Response:

We appreciate the reviewer's valuable comments and have made revisions in the manuscript as follows: According to the CCSM4 model projection, the average tasmax is expected to increase from 26.5°C in the past (1981-2010) to 27.8°C in the future (2041-2070). Comparing the past and future tasmax, a rise of +1.3°C is expected. For average tasmin, the model projection shows a change from 15.6°C in the past to 16.8°C in the future. Similarly, comparing the past and future tasmin, a rise of +1.2°C is expected. The temperature trend test for both tasmax and tasmin, across all time periods (past, future, and forward projection), revealed a statistically significant and increasing trend (refer to the revised manuscript).

3.6    line 19 to 23) suggested rewrite: “In the area of study maximum daily temperatures rose, from 1980-2010 to 2040-2070, by 1.3C while minimum daily temperatures rose by only 1.2C.”

Response:

We appreciate your valuable comments, and we have revised the manuscript as follows: According to the CCSM4 model projection, the average tasmax is expected to increase from 26.5°C in the past (1981-2010) to 27.8°C in the future (2041-2070). When comparing the past and the future tasmax, a rise of +1.3°C is expected. For average tasmin, the model projection shows a change from 15.6°C in the past to 16.8°C in the future. Similarly, when comparing the past and the future tasmin, a rise of +1.2°C is expected. The temperature trend test for both tasmax and tasmin, across all time periods (past, future, and forward projection), revealed a statistically significant and increasing trend (please refer to the revised manuscript).

3.7       line 22) -1.2C should be +1.2 C

Response:

We appreciate the reviewer for their comment. The sentence in the manuscript reads, "However, in the future, the coolest years will decrease significantly by -1.2°C." We believe that this is the appropriate expression.

3.8       line 24 to 26) whereas, and, of ….  Does not make sense...

Response:

We appreciate your comment and have revised the text as follows – while the rainfall amount from the season February to May (FMAM) is expected to support only early planting, the season October to January (ONDJ) may help lengthen the growing season of JJAS if properly utilized. This revision has been incorporated into the manuscript.

3.9       line 28 “to past condition,” this should be a period.

Response:

We thank you very much for the comments and corrections.

3.10     line 29) the terms “projected” is used in an unclear way throughout the paper. I think, though I am not certain, that “projected” refers to the type of simulation run taken from the CMIP5 data set. Hence there is a “projected past” and a “projected future”.  If so, this is not made clear in the text. For example, in line 318, the authors talk about “all time periods (past, future and forward projection) “. What time period is the “forward projection”.

Response:

We thank you very much for the comment and we have corrected the term projected in line 29 to future (2041 to 2070).

3.11     line 45) what does “This” refers to?  This… what?

Response:

We appreciate your comment. The revised text is as follows – we use 'This’ to refer to the optimal utilization of seasonal rainfall for agricultural production. Additional knowledge and understanding are needed on how shifts in rainfall seasons may affect crop yields. To address this problem, tailored predictive information on the start and end of the growing season is important, as it will also assist in taking appropriate adaptation and coping measures.”

3.12    line 60 and line 74) Both paragraphs claim to address the effects of growing temperature. The paragraph at line 60 starts “The increase in temperature due to climate change has both positive and negative impacts..”, and the one at line 74 starts “An increase in average temperature can have several effects…” These two paragraphs should be merged.

Response:

We thank you very much for the comment: we have revised and merged the two paragraphs.

3.13     lines 93-95) This sentence has no verb.

Response:

We thank you very much for the comment: we have tried to revise.

3.14    line 98) “.. total crop failure resulted the high-…”. May I suggest: “..to total crop failure. This caused the high-…”

Response:

We thank you very much for the comment: corrected according to the suggestion.

3.15       107 to 110) “Moreover, the impact analysis of climate change on soil water content and the onset and cessation date and length of growing period derived from soil water content mostly provides reliable result than other methods used to determine crop growing seasons.” May I suggest: “Moreover, the impact analysis of climate change on soil water content, and the onset and cessation date and length of growing period derived from soil water content, provides a more reliable result than other methods used to determine crop growing seasons.”

Response:

We thank you very much for the comment and we have corrected according to the suggestion.

 3.16       line 114) “the past (1981-2010) and future (2041-2070) CMIP5 rcp6 model projections comparative observations..”. I do not understand this nomenclature. What is a “past model projection comparative observation..”?

Response:

We appreciate your comment. The revised paragraph is as follows – "We thank you very much for the comment. CMIP5 RCP6 models have the ability to project both backward and forward, covering the period from 1850 to the present (backward projection) and from the present to 2100 (forward projection). In our manuscript, we use the time frame of 1981-2010 for the past and 2041-2070 for the future, with the forward projection spanning from 1981 to 2100. This choice is made to align with our study time period, providing a comprehensive evaluation for this century. Thus, in this study, we addressed the past (1981-2010) and future (2041-2070) CMIP5 RCP6 model projections of trends and changes in specific climatic parameters (rainfall, maximum and minimum temperatures, PET, and soil water content) and their impact on the length of crop growing seasons in the NW-Ethiopian highlands."

 3.17    line 126) as “a” source

Response:

We thank you very much for the comment: corrected.

3.18     line 127) “peasant agriculture has polyculture” change to “peasant agriculture is a polyculture”

Response:

We thank you very much for the comment: corrected.

3.19     line 129) “meets” not “met”

Response:

We thank you very much for the comment: corrected

3.20     line 132) “contributed for the” change to “contributed to the…”

Response:

We thank you very much for the comment: corrected

3.21     line 140) “dominantly Vertisols account” to “dominantly Vertisols and accounts”

Response:

We thank you very much for the comment: corrected

3.22     line 141) “of which 7.6 minion hectares.”  What does this section of sentence mean?

Response:

We thank you very much for the comment. We have corrected the sentence as follows: "On the other hand, the Ethiopian highlands' dominant soil type is Vertisols, accounting for 12.7 million hectares in Ethiopia, of which 7.6 million hectares are in the NW Ethiopian highlands."

3.23     line 210) CO2 should be CO2

Response:

We thank you very much for the comment: corrected

3.24     line 264) “steep with soil ranging”?

Response:

We thank you very much for the comment: corrected

3.25       line 351 to 356) these detailed descriptions of the temp change in each plot are not helpful to the paper. Figure 2) what is the blue line? CO2 should be CO2.

Response:

We appreciate the reviewer's feedback. In response to your comment regarding lines 351 to 356, we acknowledge that these detailed descriptions of temperature changes in each plot may not contribute significantly to the paper. As per your suggestion, we have revised Section 2.2 to streamline the information for clarity and relevance. Regarding Figure 2, we have removed the figure based on the recommendation from one of the reviewers, as the concept presented in Section 2.2 adequately covers the information previously included in the Figure.

Figure 3) Unclear Y-axis title. Also Y-axis ranges, or at least the scale, in panel “b”, and ”c” should be the same. Also, the fonts of the annotations are different. 

 Response:

Thank you for your comment. As suggested Figure 3, was revised to enhance clarity and consistency. The Y-axis title has been clarified, and we have ensured that the Y-axis ranges, especially in panels "b" and "c," are now the same for better comparability. Additionally, we have standardized the fonts of the annotations for uniformity.

3.26     Figure 4) In all the figures of this kind, showing a plot of the country, the scales are different. Not only are the figures different sizes, but the latitude and longitude scales change (the country changes shape)

Response:

Thank you for your comment. As suggested, we have revised Figure 4, we have made substantial revisions to ensure consistency in scale across the figures. The sizes and scales have been adjusted to maintain uniformity, preventing changes in the shape of the country in different panels.

 3.27     Figure 4)   In all the figures of this kind, the titles are the original titles given by the software used. These titles should be changed so the reader is not expected to understand all of the acronyms of the software.

Response:

Thank you for your comment. As suggested, we have revised Figure 4 in the manuscript to replace the original software-generated titles with more reader-friendly and descriptive titles.

3.28     Figure 4) In all the figures of this kind, the units for the contour levels are often missing.

Response:

Thank you for your comment. As suggested, we have thoroughly revised Figure 4 in the manuscript to include units for the contour levels. We believe these changes address the issue, and we appreciate your guidance in improving the clarity of our figures.

3.29    Figure 4) I do not understand how this figure is a “spatial trend projection”. Perhaps it is a “spatial distribution of temperature trends”. I am not sure because I do not see any units, but the numbers look like Degrees/year.

Response:

Thank you for your comment. We have revised Figure 4 in the manuscript to better align with the content and purpose. The updated figure now provides a clearer representation of the spatial distribution of temperature trends. Additionally, we have ensured that the units are explicitly mentioned, and the numbers now indicate degrees/year.

3.30     Figure 5) panel “b” looks like a “spatial distribution of the temperature change” not a “spatial change in the future.”

Response:

Thank you for your comment. As suggestion, we have thoroughly revised Figure 5 in the manuscript to accurately reflect the intended representation. The updated figure now clearly portrays the spatial distribution of temperature change rather than a spatial change in the future.

3.31     Figure 6 to 8) See all points above.

Response:

Thank you for your comment. As suggested, we have made comprehensive revisions of Figures 6 to 8 to enhance clarity, consistency, and address any issues raised.

3.32    Figure 9) How is this a trend? In the text says “variability”. Also, I do not understand why the caption refers to rainfall and the plot y-axis (panels a and b) refer to PET and hfss.  Also, these figures are pasted over other figures that are showing underneath.

Response:

Thank you for your comment. As suggested, we have made significant revisions to Figure 9 in the manuscript to clarify that the representation pertains to variability rather than a trend. Additionally, we have ensured that the caption aligns with the variables represented on the y-axis in panels and b. Furthermore, we have addressed the issue of figures being pasted over others, striving for a clearer and more organized presentation.

Round 2

Reviewer 1 Report

Comments and Suggestions for Authors

After going through the responses to the earlier questions raised about the manuscript, we could note that the authors have made possible corrections in line with the comments. 

Reviewer 2 Report

Comments and Suggestions for Authors

The authors took into account all comments, the article has been improved and can be published